**Analysis**                                                    https://doi.org/10.1038/s41559-024-02435-3

# Gaps and opportunities in modelling human influence on species distributions in the Anthropocene

**Veronica F. Frans** [1,2,3] ✉ **& Jianguo Liu** [1,2]

Understanding species distributions is a global priority for mitigating environmental pressures from human activities. Ample studies have identified key environmental (climate and habitat) predictors and the spatial scales at which they influence species distributions. However, regarding human influence, such understandings are largely lacking. Here, to advance knowledge concerning human influence on species distributions, we systematically reviewed species distribution modelling (SDM) articles and assessed current modelling efforts. We searched 12,854 articles and found only 1,429 articles using human predictors within SDMs. Collectively, these studies of >58,000 species used 2,307 unique human predictors, suggesting that in contrast to environmental predictors, there is no 'rule of thumb' for human predictor selection in SDMs. The number of human predictors used across studies also varied (usually one to four per study). Moreover, nearly half the articles projecting to future climates held human predictors constant over time, risking false optimism about the effects of human activities compared with climate change. Advances in using human predictors in SDMs are paramount for accurately informing and advancing policy, conservation, management and ecology. We show considerable gaps in including human predictors to understand current and future species distributions in the Anthropocene, opening opportunities for new inquiries. We pose 15 questions to advance ecological theory, methods and real-world applications.

Correlating species' occurrences with their surrounding habitat has been the best possible way to empirically approximate species' niches in geographic space. Species distribution models (SDMs) are statistical and machine learning tools that correlate species' locations with environmental predictors (that is, covariates, variables and parameters) to predict species' probabilities of occurrence (or occupancy, habitat suitability and presence) across geographic space and/or time[1,2]. Species–environment relationships determined from SDMs also inform on the multi-dimensional environmental gradient (hypervolume[3]) along which species' niches can be defined. Across thousands of studies and across all domains, spatial scales and taxa, this hypervolume has been commonly represented by suites of predictors relating to climate (for example, temperature and precipitation) and other abiotic interactions (altitude, latitude and topography). Such predictors have been used to estimate species distributions with high accuracy[4,5]. However, while these predictors correspond to general ecological

[1]Center for Systems Integration and Sustainability, Department of Fisheries and Wildlife, Michigan State University, East Lansing, MI, USA. [2]Ecology, Evolution, and Behavior Program, Michigan State University, East Lansing, MI, USA. [3]W. K. Kellogg Biological Station, Michigan State University, Hickory Corners, MI, USA. ✉e-mail: verofrans@gmail.com

niche requirements, the emphasis on such predictors ignores a quintessential phenomenon most relevant to the conditions of our current era: human influence.

Ample evidence has shown that human activities (or simply human presence) have direct and indirect influence on species distributions in the Anthropocene[6–10]. Such influence has been the most obvious in examples relating to human population growth[7], species invasions[11,12], urban expansion[8,13] and land-use change[14,15]. Other less obvious examples exist for species found in the most remote or well-protected environments on the globe (for example, noise pollution from increased tourism in a nature reserve for conserving giant pandas (*Ailuropoda melanoleuca*) has caused them to prefer habitats outside the reserve[16]). Despite evidence from ecological studies and international expressions of concern regarding the state of species as a result of human influence[17–20], it is unclear how often predictors relating to human activities, presence or pressures (hereafter called human predictors) are being used in SDMs.

The absence of human predictors can be especially problematic when species distributions are projected to novel environments. For example, a geographic area might be projected as suitable for a species because of its land cover and climate conditions, but is actually unsuitable due to night-time light intensity from distant residential areas[21]. If such important human predictors are not utilized, the mechanisms behind many ecological changes might not be revealed in even protected areas[22], and resources and efforts to reintroduce a species as a result of SDM predictions could be unsuccessful[23]. A similar concern exists with projecting species distributions across time based on future climate scenarios if human activities have a greater effect on species distributions than climate[24]. Thus, inadequately accounting for human predictors in species projections could largely affect broader applications or interpretations from SDMs[23], leading to false optimism about a species' future trajectory or the implementation of misinformed policies.

As SDMs are used in a wide variety of fields—from disease ecology to conservation—understanding how human predictors are currently being used in SDMs can help direct modelling efforts as human influence in the Anthropocene amplifies. In this Analysis, we conducted a systematic review to critically examine how human influence is incorporated into models of species distributions. We examined whether SDM articles acknowledged human influence and, if so, whether human predictors were incorporated in models for assessing and predicting species distributions. We compiled a list of the unique human predictors being used in SDMs so far and examined the context for their use across domains (marine, terrestrial and freshwater), spatial scales and taxa all around the globe. Acknowledging the critical intersection between biodiversity and sustainability[19,25], we also examined how these human predictors related to global Sustainable Development Goals (SDGs)[20]. Lastly, we searched for trends in model procedures for predictor selection, SDM training and forecasting, and evaluated researchers' reports on model performance.

Our synthesis demonstrates the need for advances in SDMs, as we found substantial variability in SDM studies' consideration of human influence. Since SDMs are open, easily accessible tools for conservation, management and ecological studies, covering even data-poor locations and data-deficient species, we propose that standardizing the use of human predictors in SDMs offers opportunities to (1) improve the realism and applications of predicting species distributions in novel spaces and time; (2) enhance global syntheses on the effects of human activities across various domains, taxa and spatial scales; and (3) broaden theoretical perspectives in ecology.

## The current state of human influence in SDM research

Modelling human influence (human activities, presence or pressures) on species distributions is extremely uncommon. Among 12,854 SDM

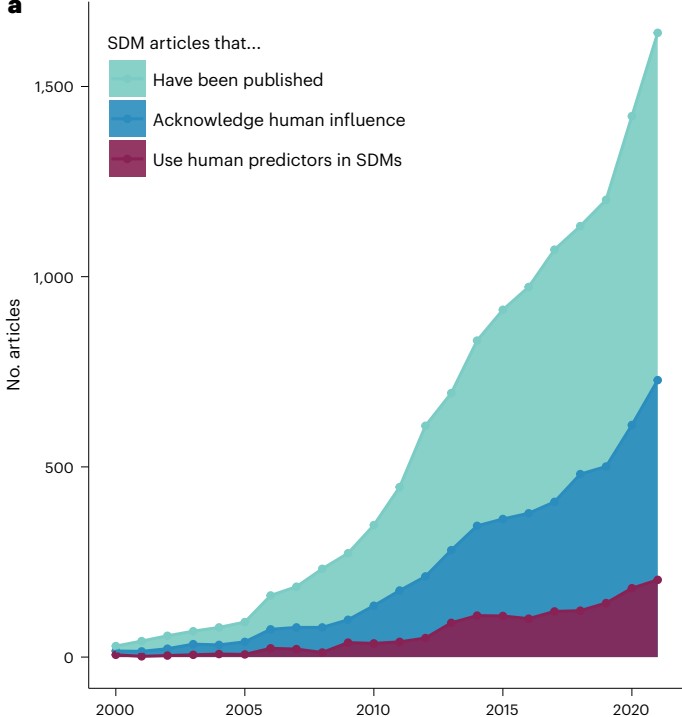

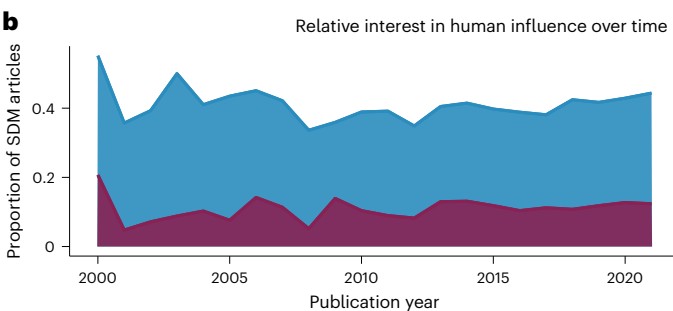

**Fig. 1 | Rarity of modelling human influence on species distributions in SDM literature. a,b**, While the number of published SDM articles acknowledging human influence on species distributions has been increasing over time (**a**, blue), the relative proportion of articles where human influence is incorporated within SDM procedures is substantially less (purple), and the interest in modelling human influence on species distributions (**b**) has plateaued to below 15% over the past two decades. These graphs represent the total articles published from 2000 to 2021 (teal), found in a Web of Science search (for search terms, see Methods). Of these, the articles that acknowledge human influence on species distributions (blue) are those that describe human influence within their abstracts. The articles that use human predictors in SDMs (purple) are those that use human predictors in SDM training for their predictions.

articles published up to 2021 and catalogued in the Web of Science (Methods and Extended Data Fig. 1), we found that 5,177 (40%) of them acknowledged human influence on species distributions within their abstracts (Fig. 1a) and only 1,429 articles published since 2000 (11%) went on to use human predictors (that is, predictors associated with human activities or human-induced pressures) within their SDMs. Another 267 articles (2%) used human predictors outside their models by, for example, masking (omitting) predicted areas of occurrence with human infrastructure or residential areas[26]. While the number of articles using human predictors in SDMs has increased over time, the relative interest in conducting such studies has plateaued to less than 15% of published SDM articles since the early 2000s (Fig. 1b).

From these 1,429 articles that used human predictors within SDMs, we found that human predictors have been used mostly in studies at

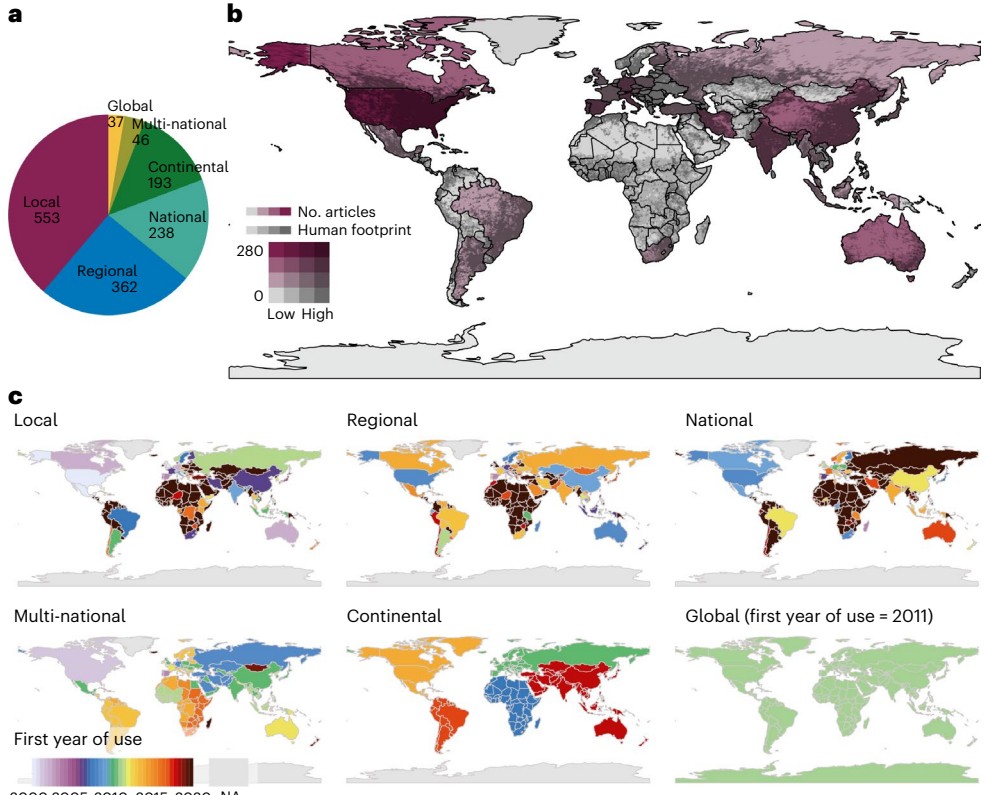

**Fig. 2 | Spatial scales, study locations and initial years of human predictor use in SDMs across the globe. a–c**, While most studies are at local, regional (within country) and national scales (**a**), there is a disparity in the global coverage of species distribution modelling studies using human predictors in model training compared with the 2020 Global Human Footprint (**b**)[28] and a temporospatial bias for when human predictors have first been used around the world across various scales (**c**). These studies represent 1,429 SDM articles published between 2000 and 2021 that include human predictors in model training. Note that the mapped studies in **b** include local to multi-national scales but exclude global and continental (all countries within-continent) scale studies; marine studies were appended to their respective countries. In **c**, we use the first years of publication between 2000 and 2021 as a proxy to signify the first year that a human predictor was used in an SDM within a given region (NA refers to locations where human predictors have not been used during this time period). See Extended Data Figs. 2, 3 and 5–8 for more detailed maps, sorted by domain, taxa, study focus and spatial scale.

local, regional (within country) and national spatial scales (Fig. 2a and Extended Data Fig. 2). Global and continental-scale studies were few (37 and 46 articles, respectively, or 3% each). While human influence is globally pervasive[27–29], most studies using human predictors in SDMs focused on the United States ($n = 274$), China ($n = 100$), Spain ($n = 100$), Italy, Germany, Iran, India, Canada, Australia, Portugal and France, totalling 931 articles (65%; Fig. 2b and Extended Data Fig. 3). In other areas, such as South America, central and southern Africa, Scandinavia, Eastern Europe and Southeast Asia, where the global human footprint is predominantly high[28], relatively few studies used human predictors in SDMs. In such areas, it was not until around 2010 that human predictors were first used in SDMs at global and continental scales. In Africa, South America and some parts of Asia especially, it was not until 2020 that human predictors were first used in SDMs at national, regional or even local scales (Fig. 2c).

Articles including human predictors in SDMs collectively modelled the distributions of over 58,000 species. These studies were not specific to domain, taxa or the focus of research (Extended Data Fig. 4). There were 1,375 terrestrial, 184 freshwater and 38 marine studies (some articles included multiple domains; Extended Data Fig. 5). Most studies were of mammals (32%), followed by birds (22%) and invertebrates (15%), and covered most of the globe (Extended Data Fig. 6). The remaining studies included herbaceous plants (11%), fish (5%), reptiles (5%), trees or shrubs (4%), amphibians (4%) and microorganisms (2%).

Studies that include human predictors primarily focused on conservation (24%), exploratory work (for example, exemplifying new methodologies or frameworks[30,31]; 23%), or species invasions (18%). Others focused on disturbance or habitat change (for example, human land-use shifts and land abandonment[31]; 15%), reintroductions or restoration (7%), food or economics (for example, food security and economically important species[32,33]; 5%), human health or safety (for example, disease vectors[34]; 5%) and human–wildlife conflict or collisions (3%) (Extended Data Fig. 4). Exploratory, disturbance or habitat change, conservation and human health or safety studies had the widest global coverage at various spatial scales, with most studies in the United States, China, France, Italy and Iran (Extended Data Fig. 7).

## Human predictor selection

We did not find any consistent patterns for the number of human predictors used in SDMs in relation to environmental (climate and/or habitat) predictors. Human predictors in SDMs ranged from as few as 1 to as many as 61, in contrast to 1–184 environmental predictors in these same studies (Fig. 3a and Supplementary Fig. 1). The mean and median number of human predictors used were three and two, respectively, compared with eleven and eight environmental predictors. Some articles exclusively used human predictors to model species distributions[35] or used more human predictors than environmental predictors[36–38]. In most cases, one to four human predictors were used with four to ten environmental predictors (Fig. 3a).

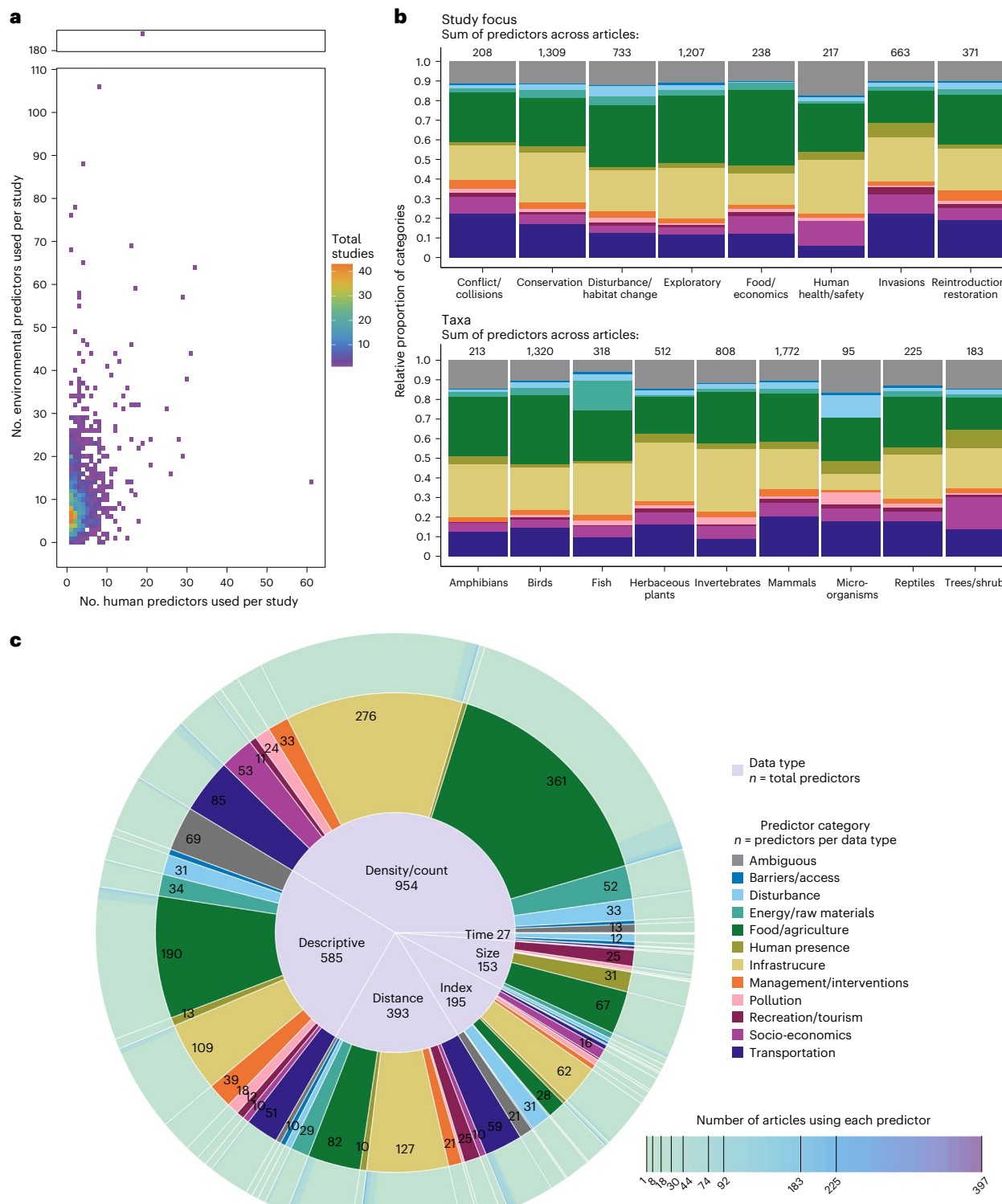

**Fig. 3 | Wide variability in human predictor use in SDMs. a–c**, There is a disproportionate use of environmental (habitat and climate) predictors compared with human predictors in SDMs (**a**), and wide variability in predictor selection across study focus and taxa (**b**), with most predictors (84%) being unique to only one article (**c**). A consistent ratio of human-to-environmental predictors for model training is not apparent from these studies. However, most studies use fewer human predictors than environmental predictors (**a**). Across taxa and areas of research focus (**b**), the majority of human predictors used pertained to food or agriculture, infrastructure, transportation or were ambiguous (that is, they could equally represent both environmental and human features). In **c**, we see a large variability in human predictor selection for SDMs. With a total of 2,307 unique human predictors used across 1,429 SDM articles,

there were six different data types (centre pie, numeric labels are counts of predictors), covering 12 different categories of human activities (middle pie, numeric labels are counts of predictors within data types; numeric labels are excluded for categories with <10 predictors). The outer pie highlights that the most commonly used predictors related to food or agriculture and infrastructure as density or count data (the coloured bars are the sums of articles using each predictor within each category and data type, with the darkest bars being the most frequently used by articles). Only 371 predictors (16%) were used by more than one article (darker bars of the outer pie). See Extended Data Table 1 for a description of data types and categories, Extended Data Fig. 8 for a map of the spatial distribution of human predictors across spatial scales and Supplementary Table 4 for a descriptive list of all predictors.

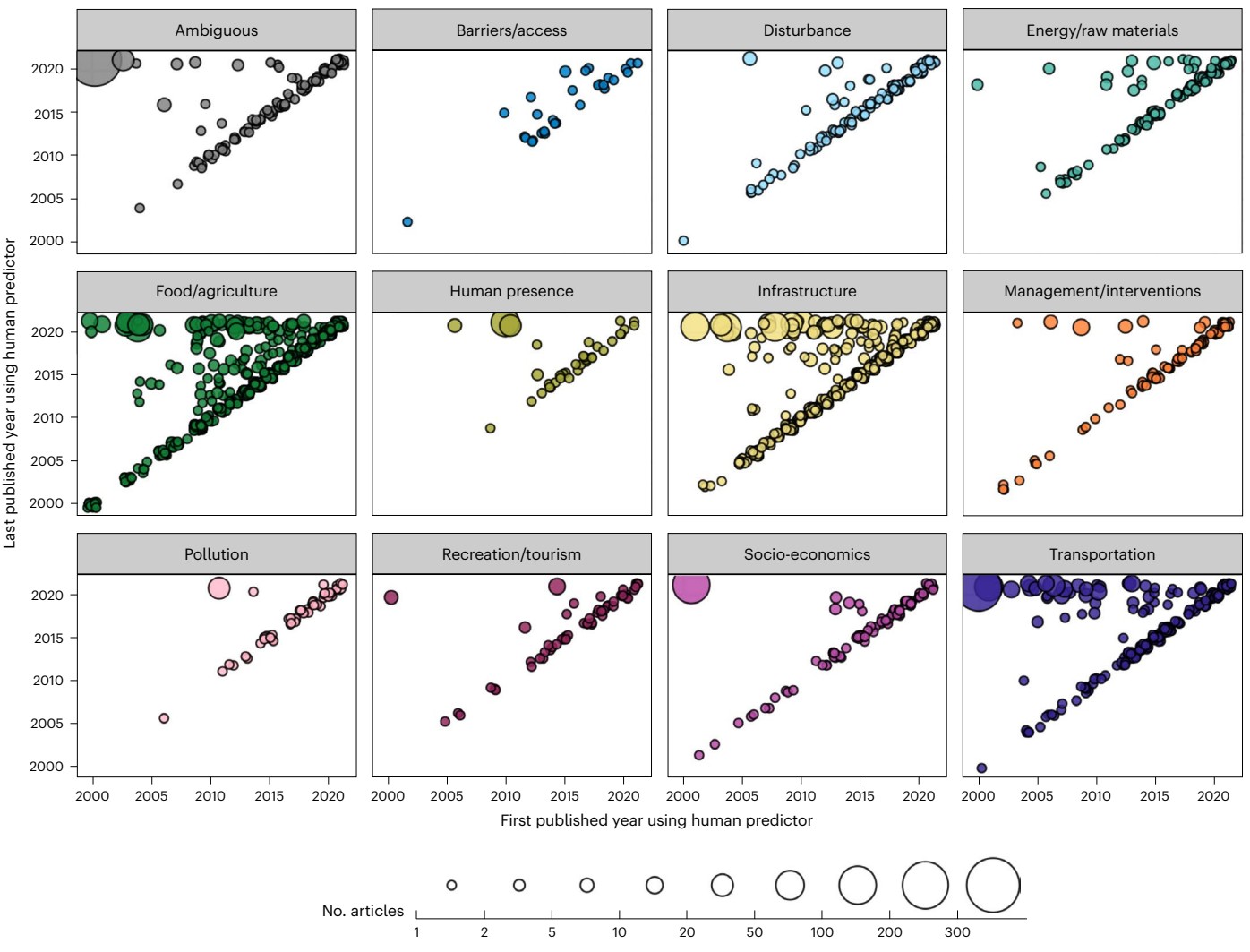

**Fig. 4 | Persistence and prevalence of human predictors since their first emergence in SDM literature.** There is a consistent emergence of new human predictors per year, but only 26% of human predictors have been used more than once over the past two decades. The persistence of a human predictor is determined by how far beyond the first published year of use a human predictor has been used in other SDM articles (*x* and *y* axes) and the prevalence of a human predictor is determined by the total number of articles in which it is used (the size of a point). The points represent the 2,307 human predictors found among the 1,429 SDM articles published between 2000 and 2021, separated by category (for category descriptions, see Extended Data Table 1).

The types of human predictors selected for SDM training were similarly variable. The 1,429 articles collectively used 2,307 unique human predictors, which is a surprisingly large number. Given the complexity of human–species interactions, we considered that human predictor selection could also depend on study context such as taxa and study focus. However, no real patterns were evident (Fig. 3b). In terms of popularity, only 16% (*n* = 371) of these predictors were used in more than one instance; most predictors (*n* = 1,936) were unique to only one article (Fig. 3c). The most common predictors were land use/land cover, distance from roads, human population density, percent agricultural areas, roads density and percent urban areas, used in 17%, 10%, 8%, 4%, 4% and 4% of articles, respectively (Supplementary Tables 4 and 6). Human footprint and human influence index were respectively used in only 74 and 36 articles (5% and 3%). Overall, human predictors ranged across many categories of human influence, with most relating to food and agriculture (*n* = 734; for example, crop area sizes, harvest intensity and commercial fishing effort), infrastructure (*n* = 617; for example, percent of buildings and intensity of development), transportation (*n* = 227; for example, distance from highways and boat traffic), energy or raw materials (*n* = 127; for example, density of powerlines and renewable energy

sites) or disturbance (*n* = 115; for example, fragmentation, logging cut-block areas and human-induced extirpation risk). Ambiguous predictors (*n* = 115) are predictors that can either represent human influence or be equally interpreted as environmental predictors (for example, land use/land cover and open areas). They were used in the SDMs of 490 articles (34%), of which 197 (14%) solely used ambiguous predictors to represent human influence[24,39,40]. New human predictors have been consistently emerging each year (Fig. 4), and their cumulative numbers vary across countries, regions, and spatial scales (Extended Data Fig. 8). The categories with the most momentum and persistence in use after first being introduced by authors or made available related to food and agriculture (*n* = 125), infrastructure (*n* = 85) and transportation (*n* = 48). We list more predictor categories in Fig. 3c, provide descriptions of all data types and categories in Extended Data Table 1 and have a full descriptive list of predictors in Supplementary Table 4.

## Potential for Sustainable Development Goal assessments

As both global biodiversity conservation initiatives and United Nations SDGs are set for multiple targets by the years 2030 and 2050[20,41],

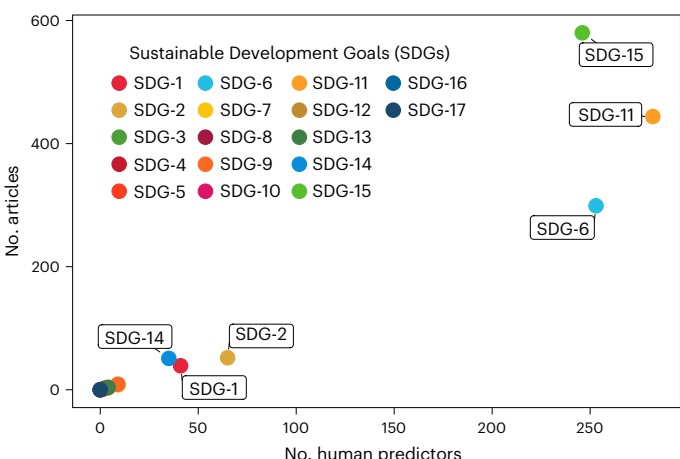

**Fig. 5 | Human predictor relationships to the 17 SDGs.** Among the United Nations' 17 SDGs, human predictors used in SDMs were most closely related to Sustainable Cities and Communities (SDG-11), Clean Water and Sanitation (SDG-6), Life on Land (SDG-15), Zero Hunger (SDG-2), No Poverty (SDG-1) and Life Below Water (SDG-14). A total of 682 human predictors relating to SDGs were used by 924 of the 1,429 articles. See Supplementary Fig. 2 for more details on article coverage and definitions for all the SDGs.

trade-offs and synergies between species and human prosperity are inevitable[25]. We thus tested whether the human predictors used for modelling species distributions related to any of the 17 SDGs. A total of 682 (30%) of them related to 13 of the 17 SDGs, modelled in 924 of the 1,429 articles (65%). These human predictors most closely related to Sustainable Cities and Communities (SDG-11, $n = 282$), Clean Water and Sanitation (SDG-6, $n = 253$) and Life on Land (SDG-15, $n = 246$). This was seen both for the number of predictors related to SDGs and the number of articles using them (Fig. 5). Other predictors found in substantially fewer articles related to Zero Hunger (SDG-2, $n = 65$), No Poverty (SDG-1, $n = 41$) and Life Below Water (SDG-14, $n = 35$). There were no predictors related to Gender Equality (SDG-5), Reduced Inequality (SDG-10), Peace and Justice Strong Institutions (SDG-16) or Partnerships for the Goals (SDG-17).

## Human predictors for forecasting and hindcasting over time

It is common for SDM studies to project species distributions not only across geographic space but also across time. However, we found that nearly half of the multi-temporal studies (past–present, present–future, past–present–future and so on) kept human predictors constant, that is, unchanged from the predictors' state at the study period (typically the present) for which the SDM was trained (136 out of 275 articles; Fig. 6). Human predictors were held constant (unchanged) for more forecasting studies ($n = 122$) than hindcasting studies ($n = 24$). The remaining articles focusing on projecting species distributions across time transformed human predictors to match the environmental predictors' past or future time frames. Human predictors that were changed across time included distances from settlements and roads (calculated as hypothetical percent changes[42]), human population sizes[24], forest or non-forested areas[39] and simulated percent habitat loss[43], among others (Supplementary Table 4). Some example human predictors that remained constant were land use or land cover[44–46], agricultural areas[47], numbers of agricultural workers[48], built-up areas[31] and human footprint index[49,50].

## Assessing SDM fit

Some articles tested and reported on the performance of using human predictors alongside environmental predictors compared with

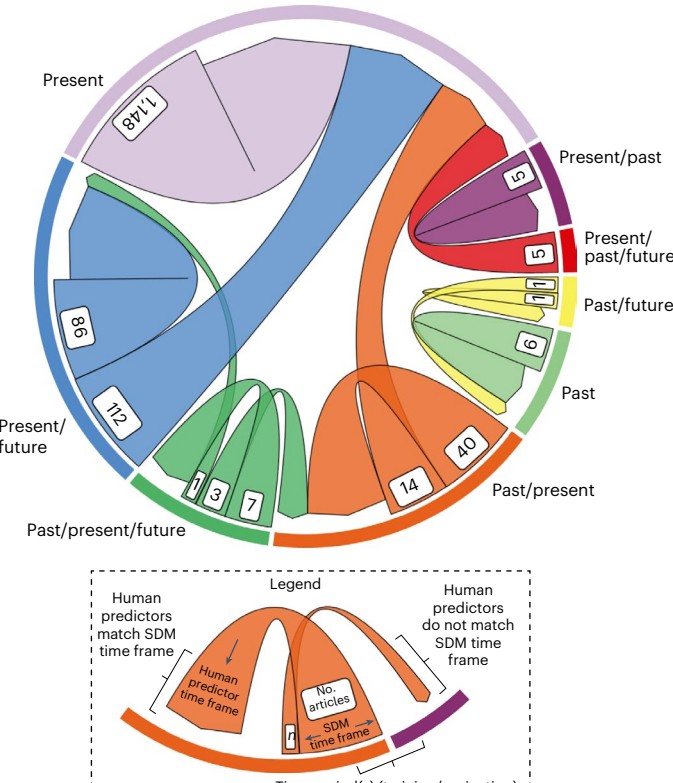

**Fig. 6 | SDM study time frames compared with human predictor time frames in model training and projection.** Most SDM articles using human predictors were both trained and projected within present time frames ($n = 1,148$), but for cases where species distributions were predicted across time (that is, hindcasting or forecasting), nearly half of the articles held human predictors constant (unchanged, $n = 136$). This disparate modelling procedure could indicate that authors either assumed that most human activities and influence would indeed remain the same in future years as far as 2050 and 2100, or that accessible human predictor data or data preparation steps for future scenarios are lacking. In this figure, the base of an arrow represents the overall time frame of the SDM for both model training and projection ($n =$ number of articles); the point of the arrow is the time frame of the human predictor used in the SDM. When an arrow folds back to its base, the overall SDM time frame (for example, present/future) matches the human predictor time frame (for example, present/future); when an arrow points away from its base and instead to another base, there is a mismatch between the study time frame (for example, present/future) and the human predictor time frame (for example, present).

environmental predictors alone, but showed no real 'rule of thumb' for human predictor selection and evaluation. SDM performance can consist of model training accuracy metrics, predictor importance, comparing predicted ranges to expert knowledge or external sources, and/or a holistic evaluation. There were 127 articles that made such comparisons (Supplementary Table 3), of which 43 stated that SDMs holistically improved when human predictors were included, while 26 stated that performance was context dependent (for example, depending on the species, scale, seasonal behaviour or preferences, or the history of a landscape)[51–53]. Another 18 articles found little to no improvement in using human predictors alongside environmental predictors, while 10 articles stated that using human predictors made SDM predictions much worse[54,55]. The remaining 30 articles did not explicitly make statements about human predictor performance. Oddly, some of the studies that found improvement in using human predictors nevertheless chose environment-only SDMs as their best models[21], while others found it essential to use human predictors in their final models—especially for future projections[24].

## New directions for SDMs in the Anthropocene

With abundant evidence of the effects of human influence on biodiversity, habitat and species abundance and distributions[18,56–58], our synthesis sets the stage for a multitude of possible directions for future research focused on understanding and predicting species distributions and niches in the Anthropocene. We propose new questions for advancing ecological theory, restructuring SDM methods and enhancing the real-world applications of SDMs.

### Advancing ecological theory

Incorporating human predictors in SDMs can further theory on how SDMs reflect ecological niches. As human predictors are increasingly made available and employed, researchers should begin to explore the following questions:

1. How will existing ecological theories and predictions on the niche, competition, disturbance and connectivity, among others, be revised when human predictors are incorporated?
2. What type of niche (fundamental, realized, Grinnellian, Hutchinsonian, Eltonian, contemporary and so on) is being modelled when human predictors are used in SDMs?
3. What are the theoretical roles of human influence on species distributions (scenopoetic/abiotic, interactive/biotic, disturbance, facilitation, mutualism, competition and so on), and will this depend on the human predictor being used or its data transformation type?
4. To what extent are human predictors correlated with environmental predictors, and when do they classify as Eltonian noise?

Various perspectives exist on the types of niches SDMs are modelling[5,59–66], and the general definition of the niche has changed through time and within ecological subdisciplines[67]. Incentives to use human predictors in SDMs would thus require re-evaluating the niche concept under these new circumstances. For example, Soberón and Nakamura[68] suggest that the type of predictor used determines whether a niche is Grinellian (SDMs using abiotic, non-interactive predictors) or Eltonian (SDMs using predictors relating to biotic interactions or resource consumption). Additionally, Moll et al.[69] propose that human interactions can be classified as super-predators, niche constructors, hyper-keystone species, risk responders and pseudo-mutualists. However, not all 2,307 human predictors used in these SDM articles may represent such Eltonian roles. Some human predictors may be interactive (for example, hunting areas, avian lead poisoning, pesticide application rates and percent protected area), while others may not (for example, artificial light intensity, human population density, settlements distance and gross domestic product). The interpretations, implications and limitations of these kinds of niches or a hybrid of them should be discussed. Methods to extract and categorize human Eltonian roles from SDMs would also need to be developed. It is also possible that if human predictors are correlated with environmental predictors due to indirect effects from human activities, the use of human predictors could be theoretically unnecessary, following the Eltonian noise hypothesis[68]. However, excluding them due to Eltonian noise could misguide the practical use of SDMs, where mechanisms could be revealed for policy and decision-making. Further investigations are needed.

Other ecological concepts also come under question with the incorporation of human influence. In connectivity analyses, for example, the inverse of SDM results are used to create resistance surfaces for informing on habitat fragmentation and important pathways or corridors for species[70,71]. When including human influence, some paradoxes may develop in connectivity concepts. For example, one study revealed that intermediate levels of habitat fragmentation could surprisingly benefit a habitat specialist[72]. Fragmentation from human influence is therefore not always a negative impact for sensitive species, but can also be positive or neutral[73,74]. These complex interactions may be difficult to generalize or anticipate, causing the need for such ecological concepts to be reinterpreted when human influence is included.

### Restructuring SDM methods

The methodological advantages and disadvantages of incorporating human predictors in SDMs should be evaluated more broadly and for each specific study based on the questions of interest. As a starting point, future research should consider the following questions as SDM methods are re-examined in the context of human influence:

1. When should human predictors be included (spatial scale, taxa, functional traits, study aims, domain, accuracy and resolution) and when are they negligible?
2. When is it necessary to consider cross-scale, local (intracoupled), distant (telecoupled) and/or adjacent (pericoupled) human predictors in SDMs?
3. What are some of the universal challenges of using human predictors when projecting species distributions into novel areas (currently unoccupied by the species) or future time frames, and how can they be addressed?
4. Despite the complexities of coupled human and natural systems, can an ontology of human predictors and standard protocols for their use be made for SDM studies?
5. What are the appropriate selection measures and data transformation types for using human predictors in SDMs?
6. How can legacy or lag effects of human influence be modelled in SDMs?
7. Which methods are appropriate for preparing current or historical human predictors for future scenarios?

Besides improving model accuracy, human predictors can enrich understandings of how human activities affect species distributions via common SDM outputs such as percent rankings of predictor importance and predictor response curves. Compared with many other ecological assessments[75,76], SDMs offer an invaluable, geographically unbiased pool of knowledge from which inferences on human activities' effects on species distributions could be synthesized; they are one of the most widespread and accessible tools in ecology, covering even data-poor locations and species. If more studies incorporate human predictors and report predictor importance and response curves, key patterns could be aggregated and summarized across domains, taxa, spatial scales and even functional traits in future meta-analytic studies. Such findings would be especially helpful for conservation-, restoration- or economically focused studies.

Clarity on appropriate protocols for selecting human predictors could expand their use in SDMs and greatly enhance the use of model outputs. Currently, numerous human predictors are being used across many contexts (Fig. 3b and Supplementary Table 4), which can make it difficult to find meaning across studies. Recent literature has called for standardizing SDM methods[4,77,78], but none specifically concerning human influence. Key human predictors need to be identified and approaches for summarizing and standardizing them are necessary for wider use. With respect to environmental predictors, a standardized suite of 30 bioclimatic predictors is already widely accepted and used by the SDM community[4,79,80], as evidenced by their use by 33% of the articles that we evaluated (Supplementary Table 5). An ontology of human predictors could be selected for use based on general improvements to SDM fit, species' responses or whether predictors correspond to human activities that are commonly considered in decision-making. To incentivize standardization, future research should focus on (1) determining the most influential human predictors on species distributions; (2) assessing whether a fixed proportion of human predictors compared to environmental predictors is appropriate, whether there is a spectrum of proportions depending on context or whether correlation with environmental predictors removes their necessity; (3) evaluating if human predictor selection is specific to taxa, domain, spatial scale, study context

and/or functional traits, and how these conditions are affected in combination with environmental predictors; and (4) creating an accessible repository of selected predictors to facilitate widespread use. Existing methods for testing the utility, importance and performance of environmental predictors in SDMs[81–85] can be expanded to include human predictors. Additionally, open data efforts such as the 'Essential Biodiversity Variables' initiative[86] could include human predictors in their considerations.

An examination of the ambiguous predictors identified in our synthesis is also needed. We questioned the status of ambiguous predictors in relation to human activities because they can represent environmental-only or human influence-only circumstances, or both. Predictors such as land cover (the most commonly used predictor across articles) and the presence or absence of certain habitat types (for example, forested or non-forested areas) may falsely represent human influence in, for example, presence-only SDMs if species' occurrences are only located in non-human-influenced areas.

Future investigations should examine the circumstances under which human predictors are necessary. Human predictors are being used in SDMs for a variety of contexts (Fig. 3b and Extended Data Fig. 4). While ample studies suggest that species' responses to human influence are scale dependent[9,13], most SDM studies used a single scale for predictor values as opposed to multiple scales, and none used human predictors that crossed scales (for example, local-scale occurrences and regional-scale predictors). Species distributions may also be affected by human activities adjacent to or distant from species' occurrence locations (pericoupling and telecoupling, respectively[87,88]), as opposed to directly within their occurrence locations (intracoupling)[89]. While we identified 393 human predictors as distance data types (for example, distance from roads or residential areas), and 409 predictors across data types were radial buffers (for example, percent agricultural areas within 4 km radius), further studies need to determine how species respond to such data compared with other data transformations. Temporal dynamics also matter where, for example, daytime and night-time distributions can vary in response to human activities[6]. Yet our synthesis revealed that few studies use temporal data types (for example, fire years and field activity periods; Fig. 3c). SDMs using human predictors to model multiple species can also expand our understanding of how human influence affects community diversity, as biotic homogenization threatens many areas around the globe[57,90]. While modelling and mapping multi-scaled and/or multi-temporal predictors to single- or multi-species occurrences may be complex, tools exist to facilitate their integration[91,92].

There is no clear trend in proper procedures for modelling human influence over time. Simply masking projections or maintaining human predictors constant through time adds a misleading weight to the impacts of climate change on species distributions and can risk misguiding managers and decision-makers concerned about human activities. Such misguidance is counterintuitive, given the multitude of studies demonstrating the magnitude of human effects on ecological communities at present[9,18,57,93]; future effects are inevitable. Human activities may be more influential on species distributions than climate—especially in predictions at shorter timescales—and human impacts could become more evident over time due to lag effects, or have lasting effects due to permanent changes to habitats or ecosystems (that is, legacy effects[9]). We thus suggest that multiple human predictor scenarios be used in projections of species distributions, similar to how climate scenarios are projected. Of course, we recognize that for some study areas, the data necessary to create human predictors for forecasting or hindcasting distributions may be limited, especially at multiple spatial scales. A lack of interdisciplinary expertise may also limit researchers in generating such predictors. One solution could be to simulate multiple potential percent increases or decreases of a predictor's values or area coverage over time[94,95] or to use propensity

score matching[96] if mechanistic predictors of human influence are unavailable. Open-access tools to simulate land-use change are also being developed[97].

## Enhancing real-world applications

Finally, considering human predictors is paramount for advancing the real-world applications of SDMs. We pose the following questions for applications-focused research:

1. How does the inclusion of human predictors in SDMs affect the way protected areas are defined and evaluated?
2. How can human predictors in SDMs affect evaluations of conservation or management progress?
3. Which human predictors are the most helpful for identifying ecological sinks or traps?
4. Which human predictors would best represent linkages between SDG progress and species distributions over time—especially beyond SDG-13 (Climate Action), SDG-14 (Life below Water) and SDG-15 (Life on Land)?

SDMs are commonly used to map the ranges of species of concern, define protected areas, highlight areas of potential human–wildlife conflict and enhance the genetic connectivity and diversity of populations, among others. SDMs are also used to track changes in species' ranges over time, especially under climatic or anthropogenic pressures. These uses inform local, regional, national and even international incentives and policies regarding biodiversity protection. With human influence perforating most landscapes and seascapes either directly or indirectly[28,29,98], current gaps in using human predictors in SDMs risk missing important opportunities for conservation and management practices. It is especially important to consider human predictors for future projections to avoid the misallocation of resources or missteps in climate mitigation. Evaluating SDM projections with and without human predictors can also assist in identifying and mapping ecological traps or sinks for critical species[99].

Around the globe, protected areas have a range in human presence[22,100,101]—from complete absence to domination—but the current trend of SDMs (that is, using environmental predictors only) risks biasing how current and future protected areas are being defined. This is particularly important as the world is promoting the global '30 × 30 Initiative' to triple the size of protected areas to 30% of Earth's lands and oceans by 2030[41,102] while also trying to achieve major SDGs[20]. SDMs for defining protected areas can employ human predictors to assess potential spectra of human influence to find balances between conservation, development and sustainability. While SDG indicators directly relating to species distributions have already been identified under SDG-14 (Life below Water) and SDG-15 (Life on Land), studies are continually emerging that show that species within protected areas are linked to other SDGs, such as Decent Work and Economic Growth (SDG-8, tourism increasing the income around protected areas), Industry, Innovation and Infrastructure (SDG-9, building roads around protected areas for access) and even Partnerships for the Goals (SDG-17, international conservation breeding programmes introducing individuals to new locations)[25]. Beyond protected areas, even human predictors pertaining to Peace, Justice and Strong Institutions (SDG-16) could correlate with species distributions, as issues such as systemic racism in urban areas can impact biodiversity at national scales[103]. An assessment of species distribution changes over time in relation to the United Nations' 231 SDG indicators and across multiple taxa may reveal the relevance of species to all sectors of global policy and human flourishing.

Incorporating human predictors in SDMs may also change how conservation and management progress is traditionally evaluated. For example, supplementary tools for SDMs, such as multivariate similarity surfaces and limiting factor mapping, can highlight locations where habitat suitability is compromised and which predictors compromised them[23,104]. Accessible protocols for interpreting human

predictor importance or responses should be developed for managers and decision-makers, as well.

## Conclusions

As ecosystems continue to transform from natural systems to increasingly coupled human–natural systems[9,105], and species distributions continue to shift in response to changing climate and increasing human activities, methodological advances offer promise for developing new and revising existing ecological theories. A species' niche is generically defined by biotic and abiotic interactions, but our current era, the Anthropocene, adds further complexities due to human influence. As SDMs are powerful, easily accessible tools used for a variety of study aims across domains, taxa and spatial scales, they can provide much-needed information to ensure species persistence under impending climate change and rising human populations and activities worldwide. Further research to advance the incorporation of human predictors in SDMs is needed to enhance their applications and ensure ecological sustainability.

## Methods

### Literature search

We used the Web of Science to search its Core Collection for all SDM articles published through 31 December 2021, using search terms that were general and synonymous to SDMs, as described in Franklin[1] (search string: *TS = (('SDM\*' OR 'environmental niche model\*' OR 'species niche model\*' OR 'bioclimatic niche model\*' OR 'habitat suitability model\*' OR 'ecological niche model\*' OR 'habitat model\*')) AND DT = (Article) AND PY = (1900–2021)*, where TS is 'Topic', DT is 'Document Type', and PY is 'Year Published'). This yielded 12,854 articles. While we acknowledge that more articles could have been captured using additional search terms (for example, listing SDM algorithms), a test using terms such as 'occupancy model', 'resource selection function\*' or 'niche model\*' showed that our choice of general search terms and their resulting articles were sufficient to capture the current state of modelling human influence on species distributions. Following the Preferred Reporting Items for Systematic reviews and Meta-Analyses (PRISMA) framework[106] (Extended Data Fig. 1), we screened 12,683 of these articles' abstracts to identify articles acknowledging or describing human influence on species distributions, using the 'revtools' package[107] in R[108]. Given the large number of articles, we ensured transparency and replicability of the abstract screening process by developing a dictionary of terms related to human influence (that is, a list of words or phrases used by authors that caused us to accept papers, along with synonyms based on those terms; Supplementary Table 1). This abstract screening approach is similar to Pham et al.[109], except we did not use machine learning. We manually reviewed ~300 abstracts at a time, added terms to this dictionary and then searched along the entire pool of abstracts to accept articles based on the updated terms. We repeated this for 28 iterations, allowing us to manually screen all rejected article abstracts (*n* = 7,506), manually accept 551 article abstracts, automatically accept 4,626 article abstracts from the 477 terms added to the search and manually review a total of 5,177 full articles and their supplementary materials (Extended Data Fig. 1).

In the full-article screening, eligible articles were those that used traditional, correlative SDMs to model species distributions (as opposed to expert-opinion-based or deductive habitat suitability models) and included human predictors in SDM training. Human predictors, also known as anthropogenic predictors, are those that include an indicator of human activities, presence or pressures. These include predictors that directly allude to human influence (for example, human population size, human footprint, distance from residential areas) or indirectly allude to human influence (for example, protected versus unprotected areas and land use/land cover). We also noted articles using human predictors outside SDMs (for example, by masking predictions or highlighting areas of concern) but did not use them in the rest of our study. Any rejected articles were marked for one of three of the following reasons: (1) the article did not use a traditional, correlative SDM for modelling species distributions (for example, species abundance or density models or deductive, expert opinion models are rejected; for lists of typical SDM algorithms that are accepted, see Supplementary Table 3 and Extended Data Fig. 9); (2) no human predictors were used in SDM model training (that is, no human predictors in the paper, or human predictors were used as masks, detection probability estimates or in a post-analysis of an SDM); or (3) it was not a research article on modelling species distributions (for example, a book chapter, literature review or a model of disease, fire, cover or virtual species) or the authors used SDMs from another source. To better align our analysis with the start of global data initiatives[110–112], we later chose to remove articles before the year 2000 (*n* = 74). Of the full articles, we accepted 1,429 (13 were unavailable) and reviewed their full text and supplementary materials for synthesis.

### Systematic review and synthesis

We catalogued information from each of the 1,429 full articles identified as relevant in the full-text screening (for full reference list, see Supplementary Information). This information included the general focus (or aim) of the study (as stated by the authors in the abstract or introduction), spatial scale of the study area, study area countries, the study's time frame (past, present and/or future SDM training and projection), the time frame represented by human predictors (including simulated scenarios across time), the taxa studied, study domain (terrestrial, marine or freshwater habitat type) and SDM algorithms. For each article, we also listed the human predictors' names and the total numbers of environmental predictors used in the SDMs. We provide a description of these data in Supplementary Table 2, corresponding to Supplementary Table 5.

We synthesized the catalogued data entries using R v.4.3.0 (ref. 108). To determine the distribution of studies compared with human influence, we mapped the numbers of studies in each country against a gridded 2020 Human Footprint Index[28]. We summed the numbers of articles covering various domains, taxa and a range of general study aims, and mapped their global coverage as well. The maps were made using the 'tmap' R package[113] and ArcPro v.3.1 (ref. 114). We compared the numbers of human predictors with environmental predictors used in SDMs by creating a density plot of the frequency of articles modelling each human-to-environmental predictor ratio.

We simplified our list of predictors, as named by authors, to synthesize similar predictor names across articles. We identified transformations of predictor data (for example, percent or distance data types or various units) as unique predictors to simulate their treatment by the SDM articles' authors (for example, cropland areas and cropland percent may be used in the same SDM of a study). We then sorted the predictors by first assessing their data type (Extended Data Table 1). We defined data types as (1) density/count, for predictors relating to sums or frequencies of human activities (for example, road density and household income); (2) descriptive, for predictors that are typically categorical (factors; for example, presence/absence of barriers and land cover types); (3) distance, for predictors measuring distance from, for example, human infrastructure or locations of human activities; (4) index, for predictors calculated from a combination of other predictors (for example, human footprint); (5) size, for predictors describing the length, width, height or area of an object of human influence (for example, building height and road length); and (6) time, for predictors relating to the temporal occurrence of a human activity (for example, period of field activities[115] or prescribed fire years[116]).

We then assigned the synthesized human predictors to 1 of 12 categories of human influence (Extended Data Table 1): (1) barriers/access, for predictors describing the facilitation or deterrence of movement (for example, fence presence/absence and passable/impassable stream barriers); (2) disturbance, for predictors describing, for

example, habitat fragmentation, deforestation, degradation, change in naturalness, or indices of disturbance or avoidance (for example, human perturbation index and marine human impact); (3) energy/ raw materials, for predictors relating to energy infrastructure (for example, wind farm distance, dams density and renewable energy lease sites) or extractions of fuels or other materials (for example, oil well pads, seismic lines, dredging and disposal areas, historic mines and mine distance); (4) food/agriculture, for predictors describing the cultivation or harvest of food products (for example, percent farmlands or their distance, livestock or cultivated product density or abundance, livestock encounter rates, harvest intensity and fishing); (5) human presence, for predictors that are derived from multiple features related to humans, and that are typically synthesized into indices or intensities (for example, anthropogenic biome, high/low human activity, human footprint, human influence index and human features distance); (6) infrastructure, for predictors describing developed areas (for example, urban or residential areas, building types, housing, land ownership and military training areas); (7) management/ interventions, for predictors relating to protection, conservation or management actions or locations (for example, protected area distance, non-hunting area distance and reintroduction site nuclei); (8) pollution, for predictors describing chemical, noise or light pollution or intensity (for example, night or artificial light intensity) or effects from pollutants (for example, count of poisoning incidents); (9) recreation/tourism, for predictors relating to, for example, trails, hunting pressure, or scenic locations; (10) socio-economics, for predictors describing human population sizes or densities, demographic and social structures (for example, human poverty, education, types of water access), jurisdiction (for example, state names), illegal activities (for example, opium eradication areas) and finances (for example, gross domestic product and household income); (11) transportation, for predictors typically relating to human movement or the movement of goods (for example, roads density or distance and shipping intensity); and (12) ambiguous, for predictors that can equally represent environmental predictors (for example, land use/land cover or forested/unforested areas).

We extracted the first and last (most recent) years of human predictor use to examine the persistence and prevalence of human predictors being used in SDMs over the years. We used years of publication as a proxy for the years when each predictor was used. We plotted these sets of years per predictor as scatter plots, faceted by the 12 predictor categories. We mapped the first years of human predictor use in each study area across local, regional, national, multi-national, continental and global scales. We also mapped the total number of unique human predictors used across these spatial scales.

From this list of predictors, we used the 'text2sdg' R package[117] to mine the predictor names and assign SDGs to them, where appropriate. We calculated the sum of SDGs per predictor and plotted them using code adapted from the 'SDGDetector' package[118].

After renaming and categorization, this list of predictors was exported as a table, with data types, data categories, predictor names, study time frames, modelled taxa, study focus, number of articles, SDGs, number of SDGs and corresponding article identification numbers for each predictor. This dataset is provided here as Supplementary Table 6. From it, we calculated the sum of unique predictors used across each study focus and taxonomic group, and the frequency of predictors across articles, data types and categories.

Finally, among these articles, we also looked for author statements that holistically (both quantitatively and/or qualitatively) evaluated the performance of SDMs with human predictors compared with SDMs using only environmental (habitat and/or climate) predictors. These statements were found in the results and/or discussion sections of the articles that used both model schemes. The authors' evaluations could be based on SDM performance measures (for example, accuracy, predictor importance or statistical significance), model selection

procedures (for example, step selection), differences in predictions (for example, ranges and extents) and/or support from literature or expert knowledge. We used a vote counting method, simply recording the number of such articles stating that SDMs performed (1) better when including human predictors, (2) worse or (3) no difference was found, or that (4) performance depended on multiple other factors (for example, differences depending on scale, resolution or modelled species), so it could not be strictly determined, or (5) comparable model schemes were done, but the authors did not discuss performance. We summed these five types of conclusions to determine overall trends in SDM performance.

### Reporting summary

Further information on research design is available in the Nature Portfolio Reporting Summary linked to this article.

### Data availability

The datasets developed from this study are available as Supplementary Tables 5 and 6. They are also available on the Figshare repository (https://doi.org/10.6084/m9.figshare.24225316)[119]. Source data are provided with this paper.

### Code availability

The code used for this study were made using R version 4.3.0 and is available on the Figshare repository (https://doi.org/10.6084/m9.figshare.24225316)[119] and GitHub (https://github.com/vffrans/Human_influence_SDMs).

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

## Acknowledgements

We thank E. Zipkin, C. Klausmeier, T. Koffel, L. Schmitt Olabisi, M. Leibold and the Leibold Lab for comments on earlier versions of this manuscript. V.F.F. was supported by the National Science Foundation Graduate Research Fellowship (fellow ID: 2018253044), Michigan State University Enrichment Fellowship, Harvey Fellowship and the National Science Foundation Long-term Ecological Research Program (DEB 2224712) at the Kellogg Biological Station (KBS contribution no. 2378). We are grateful for additional support from the National Science Foundation (grant nos. 1924111, 2033507 and 2118329) and Michigan AgBioResearch (received by J.L.).

## Author contributions

V.F.F. and J.L. conceived the ideas for this study. V.F.F. designed the methodology and collected, reviewed and analysed the literature and data. V.F.F. evaluated the results, with support from J.L. V.F.F. drafted the manuscript, with critical revisions by J.L. Both authors gave final approval for publication.

## Competing interests

The authors declare no competing interests.

## Additional information

**Extended data** is available for this paper at https://doi.org/10.1038/s41559-024-02435-3.

**Correspondence and requests for materials** should be addressed to Veronica F. Frans.

**Extended Data Table 1 | Data types and categories assigned to human predictors listed in the accepted SDM articles of the systematic review**

| | Definition | Example predictors |
|---|---|---|
| **Data types** | | |
| density/count | predictors relating to sums or frequencies of human activities | road density, household income, avian pesticide deaths |
| descriptive | predictors that are typically categorical (factors) | presence/absence of barriers, land cover types |
| distance | predictors measuring distance from human-related structures, land cover, or activities | distance to trails or roads, distance from non-hunting reserves |
| index | predictors calculated from a combination of other predictors | human footprint, human influence index, human activity levels |
| size | predictors describing the length, width, height, or area of an object of human influence | building height, road length |
| time | predictors relating to the temporal occurrence of a human activity | period of field activities, or prescribed fire years, mean annual inundation time, year moved into housing |
| **Categories** | | |
| ambiguous | predictors that can be equally representative of environmental predictors | land use/land cover or forested/unforested areas |
| barriers/access | predictors describing the facilitation or deterrence of movement | fence presence/absence, passable/impassable stream barriers |
| disturbance | predictors describing habitat fragmentation, deforestation, degradation, change in naturalness, or indices of disturbance or avoidance | human perturbation index, marine human impact, logging cut-block area, harvested forest percent, logging duration |
| energy/raw materials | predictors relating to energy infrastructure or extractions of fuels or other materials | wind farm distance, dam density, renewable energy lease sites, oil well pads, seismic lines, dredging and disposal areas, historic mines, mine distance |
| food/agriculture | predictors describing the cultivation or harvest of food products | percent farmlands or their distance, livestock or cultivated product density or abundance, livestock encounter rates, harvest intensity, fishing |
| human presence | predictors that are derived from multiple features related to humans, and that are typically synthesized into indices or intensities | anthropogenic biome, high/low human activity, human footprint, human influence index, human features distance |
| infrastructure | predictors describing developed areas | urban or residential areas, building types, housing, land ownership, military training areas |
| management/interventions | predictors relating to protection, conservation, or management actions or locations | protected area distance, non-hunting area distance, reintroduction site nuclei |
| pollution | predictors describing chemical, noise, or light pollution, or intensity or effects from pollutants | night or artificial light intensity, count of poisoning incidents |
| recreation/tourism | predictors relating to recreational trails, hunting, tourism, or scenic locations | recreational areas, distance to ski resorts, trails index, parks, hunting registration |
| socio-economics | predictors describing human population sizes or densities, demographic and social structures, jurisdiction, illegal activities, and finances | human poverty, education, types of water access, state names, opium eradication areas, gross domestic product, household income |
| transportation | predictors typically relating to human movement or the movement of goods | roads density or distance, shipping intensity |

Data types refer to the format of the human predictor, while data categories refer to kind of human influence (human activities, presence, or pressures) that a human predictor represents.

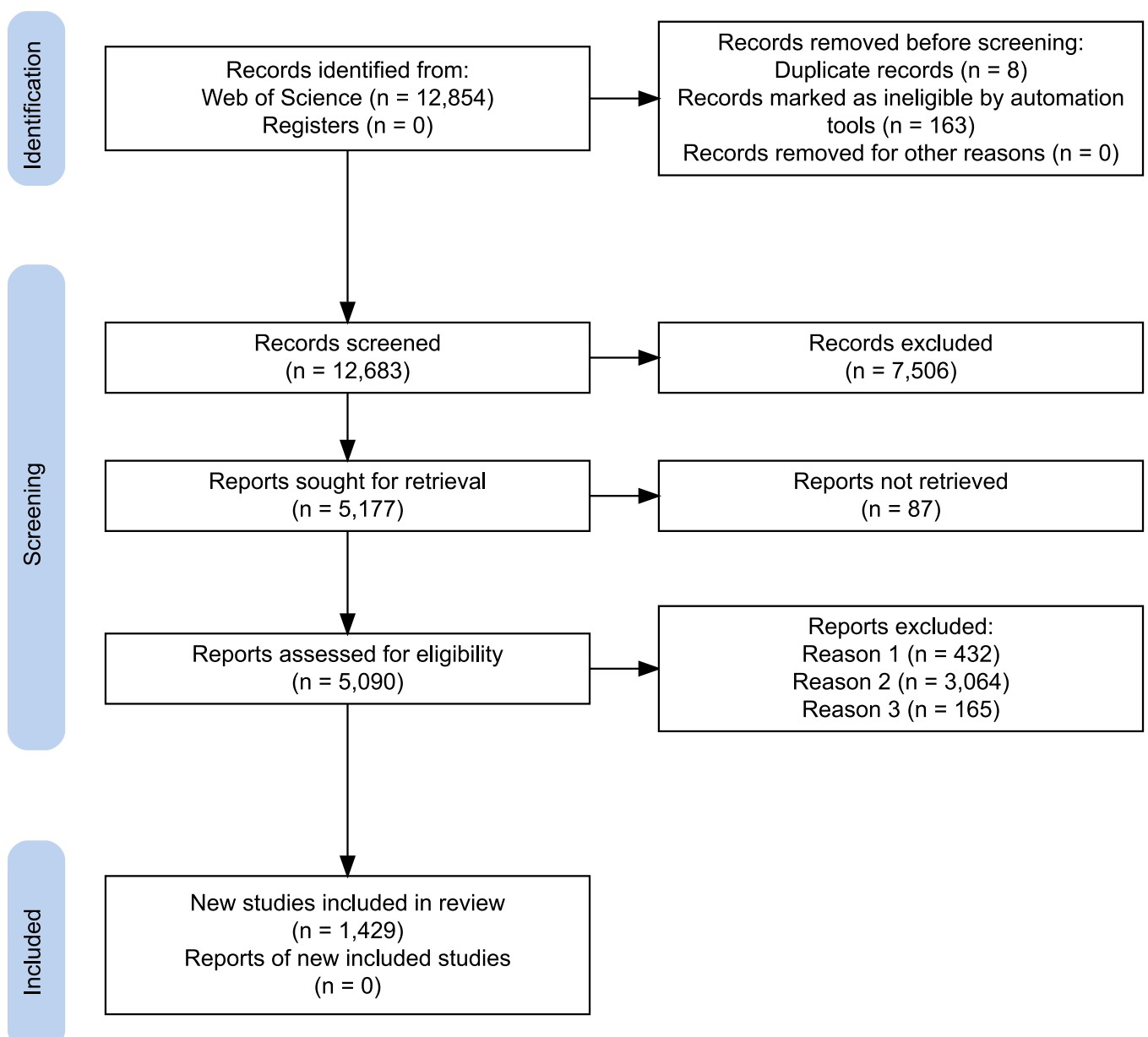

**Extended Data Fig. 1 | PRISMA workflow for article search, screening, selection and inclusion in the literature review and synthesis on human predictor use in SDMs.** Using Web of Science, we found 12,854 articles under the search string, *TS = (('species distribution model*' OR 'environmental niche model*' OR 'species niche model*' OR 'bioclimatic niche model*' OR 'habitat suitability model*' OR 'ecological niche model*' OR 'habitat model*')) AND DT = (Article) AND PY = (1900–2021)*. Of these articles, there were 8 duplicates and 163 articles published after 2021 that were removed using automation tools (R coding). 12,683 article abstracts were screened (see Table S1 for abstract screening procedure), of which 5,177 mentioned human influence on species distributions and were thus accepted. From those abstracts, 5,090 full articles were accessible and reviewed, assessing whether human predictors were used in SDM training. Of these articles, a total of 3,661 were rejected for the following reasons: Reason 1: a traditional, correlative SDM was not used for modelling species distributions (see list of typical algorithms in Table S2 and Extended Data Fig. 9); Reason 2: no human predictors were used in SDM model training (that is, no human predictors in the paper, or human predictors are used as masks or in a post-analysis of an SDM); and Reason 3: not a research article (for example, a book chapter, literature review), or the authors used SDMs from another source. This yielded a final total of 1,429 accepted articles for our synthesis. Note that the term 'records' under the PRISMA framework refers to 'abstracts' in our case, and 'reports' and 'studies' both refer to 'full articles'.

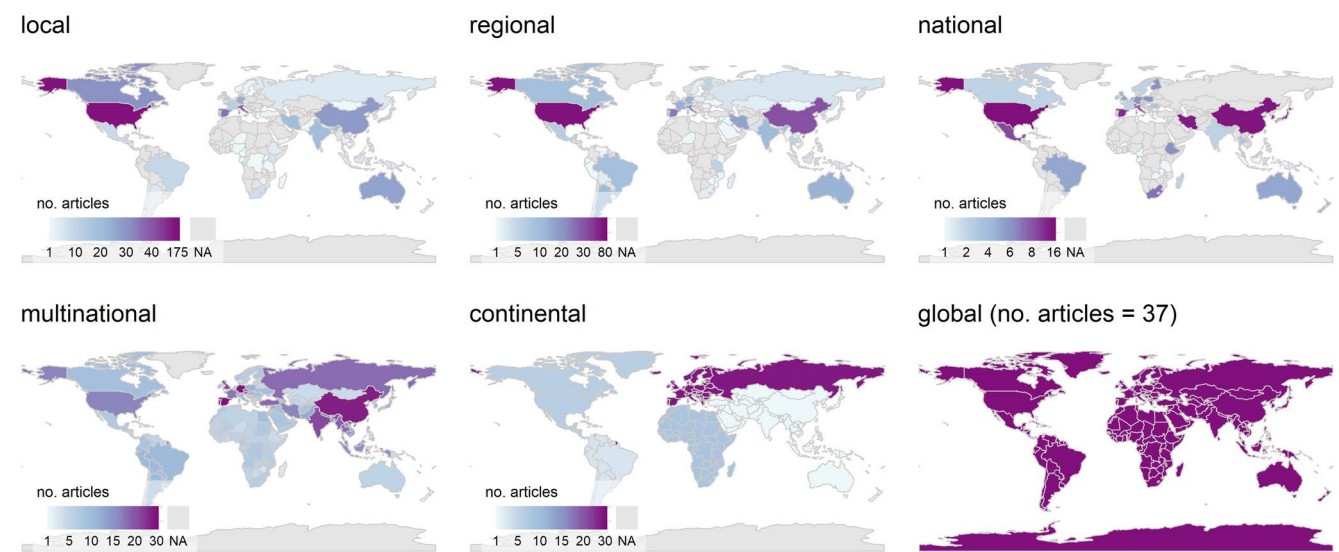

**Extended Data Fig. 2 | Spatial distribution of articles using human predictors in SDMs, based on the spatial scale of each study.** These represent 1,429 SDM articles published from 2000 to 2021. Note that marine articles are appended to their respective countries.

## study areas at local, regional, and national scales

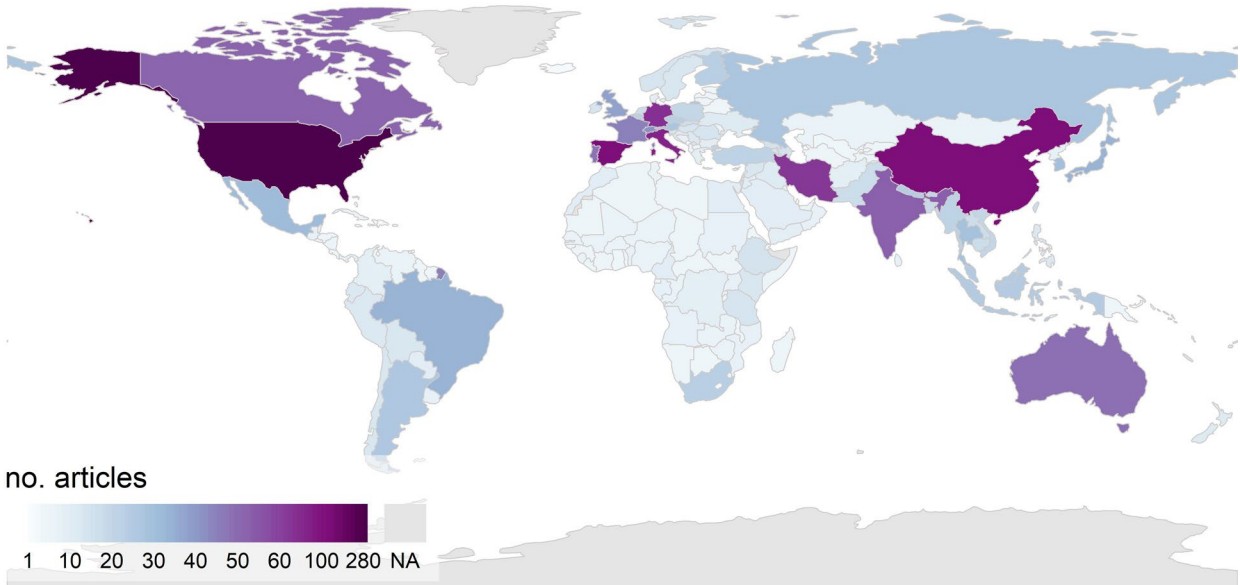

**Extended Data Fig. 3 | Spatial distribution of articles using human predictors in SDMs.** These represent 1,429 SDM articles published from 2000 to 2021. Note that marine articles are appended to their respective countries, and continental and global-scale studies are excluded.

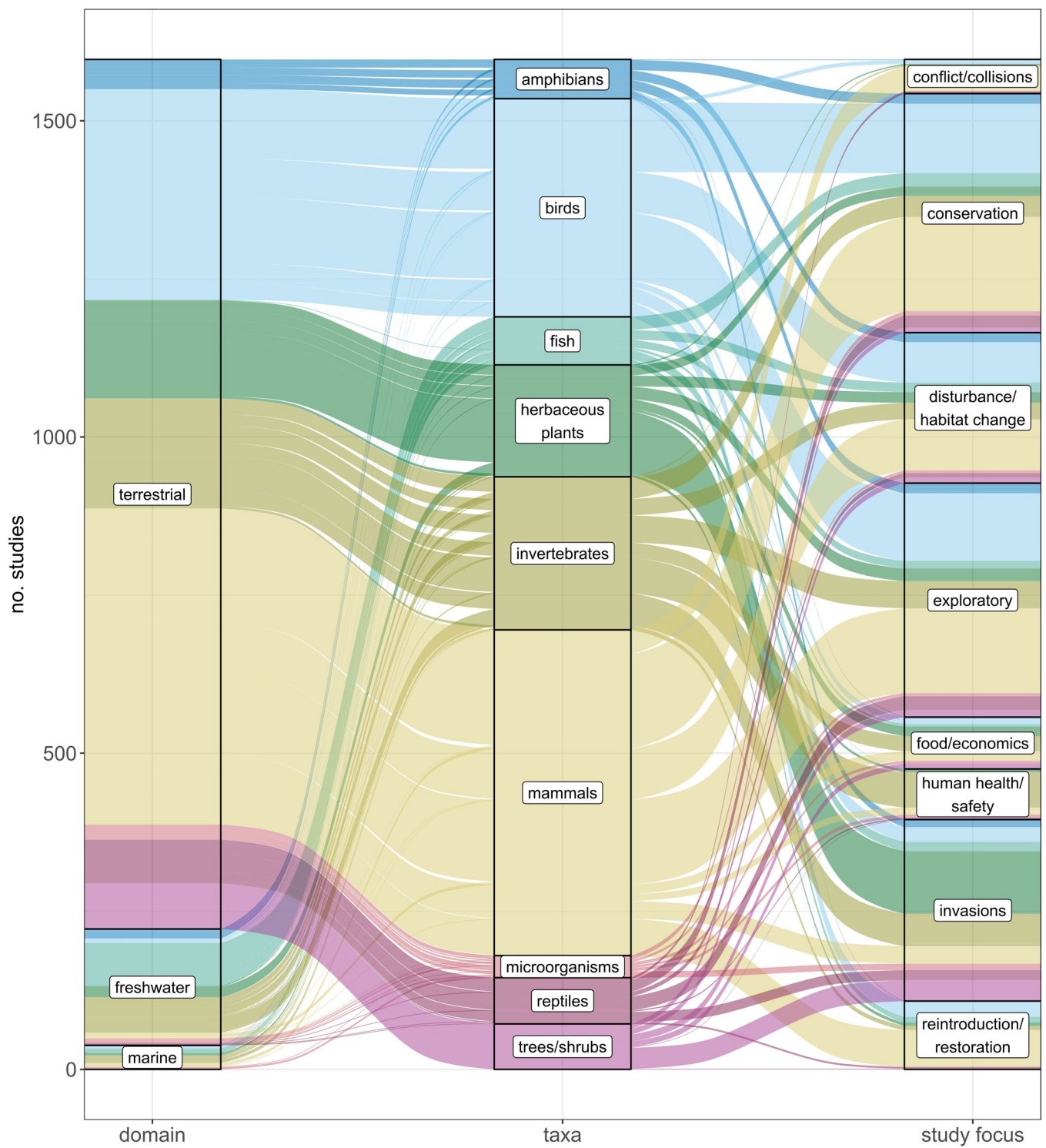

**Extended Data Fig. 4 | Range in the domains, taxa and focus (aims) of studies being conducted that include human predictors in SDMs.** Note that the total number of studies here exceeds the total number of full articles in the synthesis (n = 1,429), since some articles covered multiple domains and taxa. Maps showing the global distribution of articles across domain, taxa and study focus are located in Extended Data Figs. 5–7.

## freshwater

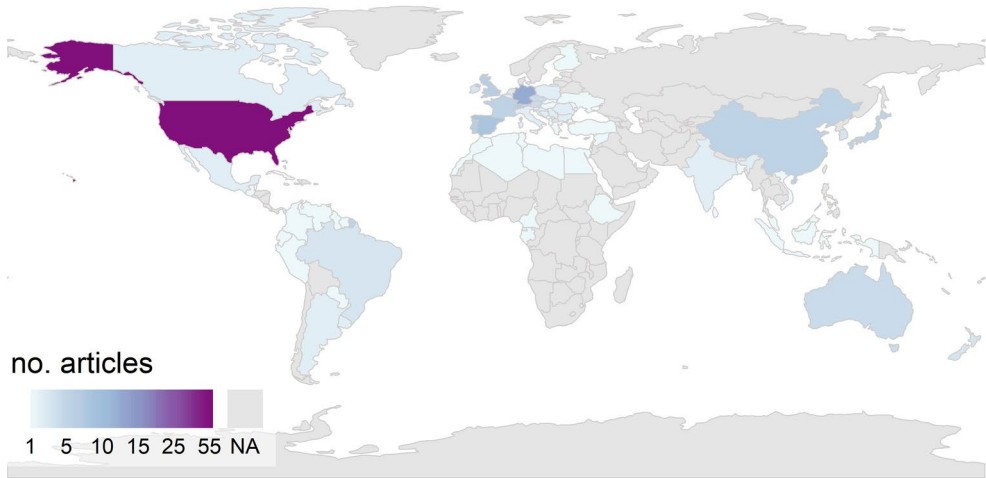

## marine

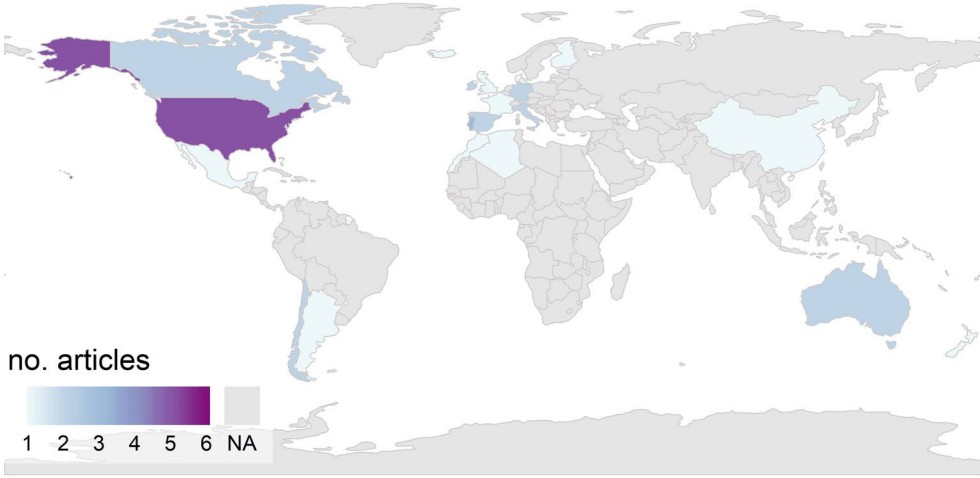

## terrestrial

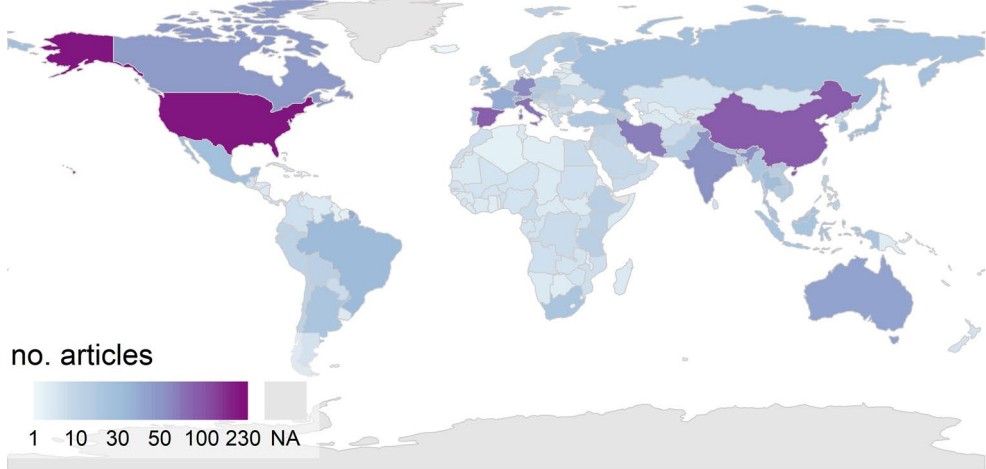

**Extended Data Fig. 5 | Spatial distribution of articles using human predictors in SDMs in freshwater, marine and terrestrial domains.** These represent 1,429 SDM articles published from 2000 to 2021. Note that marine articles are appended to their respective countries, and continental and global-scale studies are excluded.

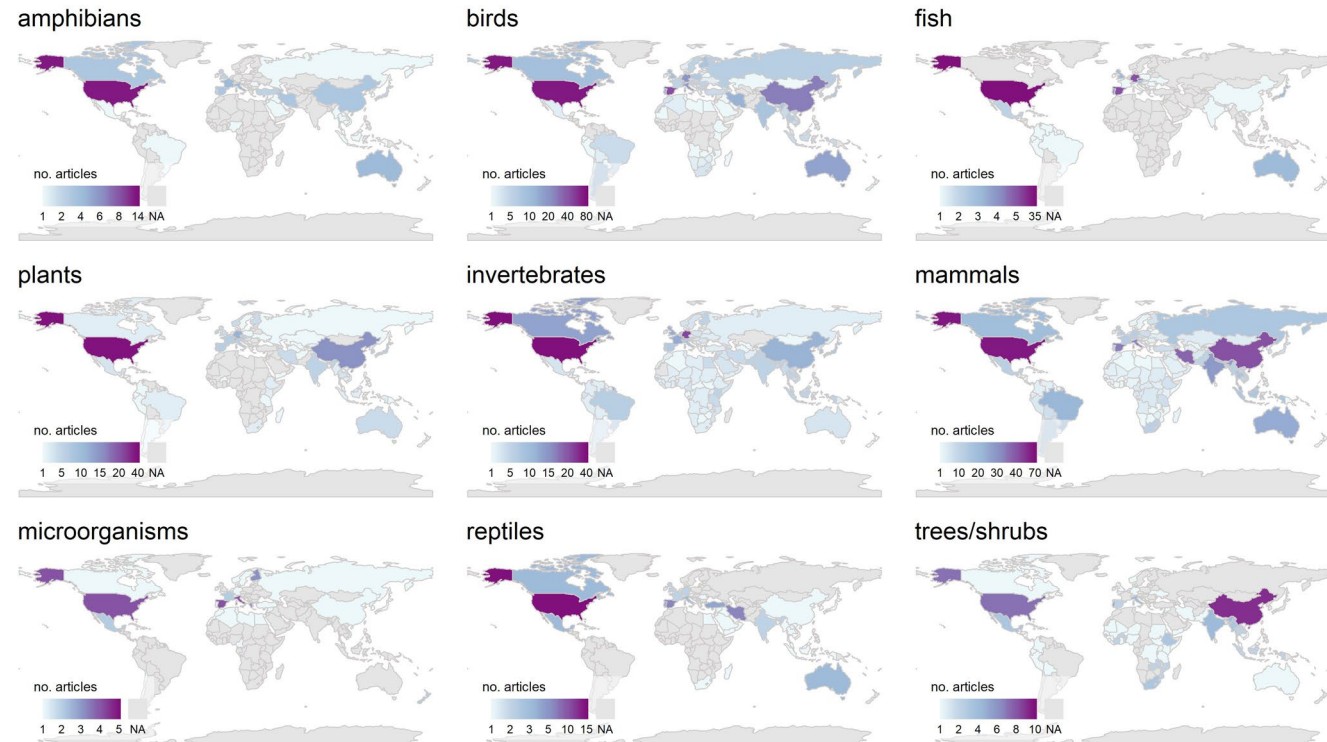

**Extended Data Fig. 6 | Spatial distribution of articles using human predictors in SDMs for various taxa.** These represent 1,429 SDM articles published from 2000 to 2021. Note that marine articles are appended to their respective countries, and continental and global-scale studies are excluded. At local to multi-national spatial scales, these maps collectively represent over 44,000 species.

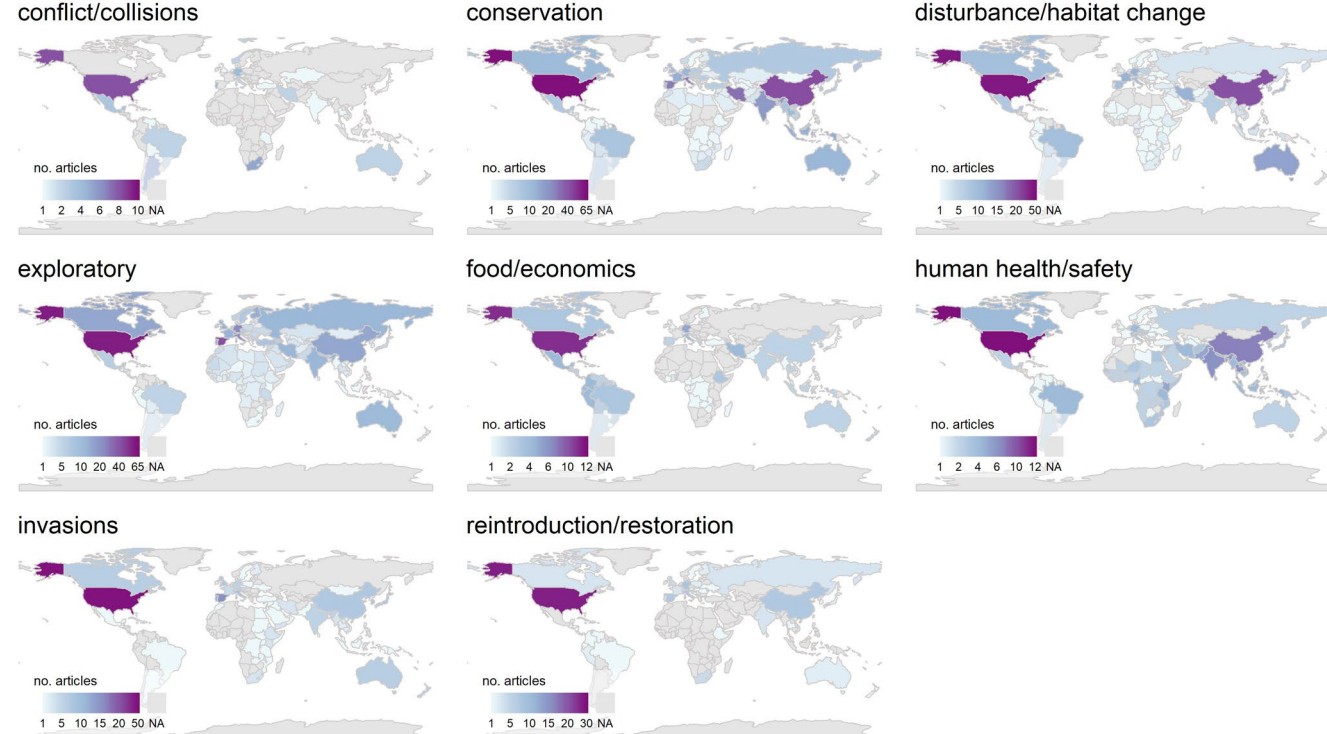

**Extended Data Fig. 7 | Spatial distribution of articles using human predictors in SDMs, based on the stated focus of the study, ranging from topics such as conservation and disturbance to economics and human health.** These represent 1,429 SDM articles published from 2000 to 2021. Note that marine articles are appended to their respective countries, and continental and global-scale studies are excluded.

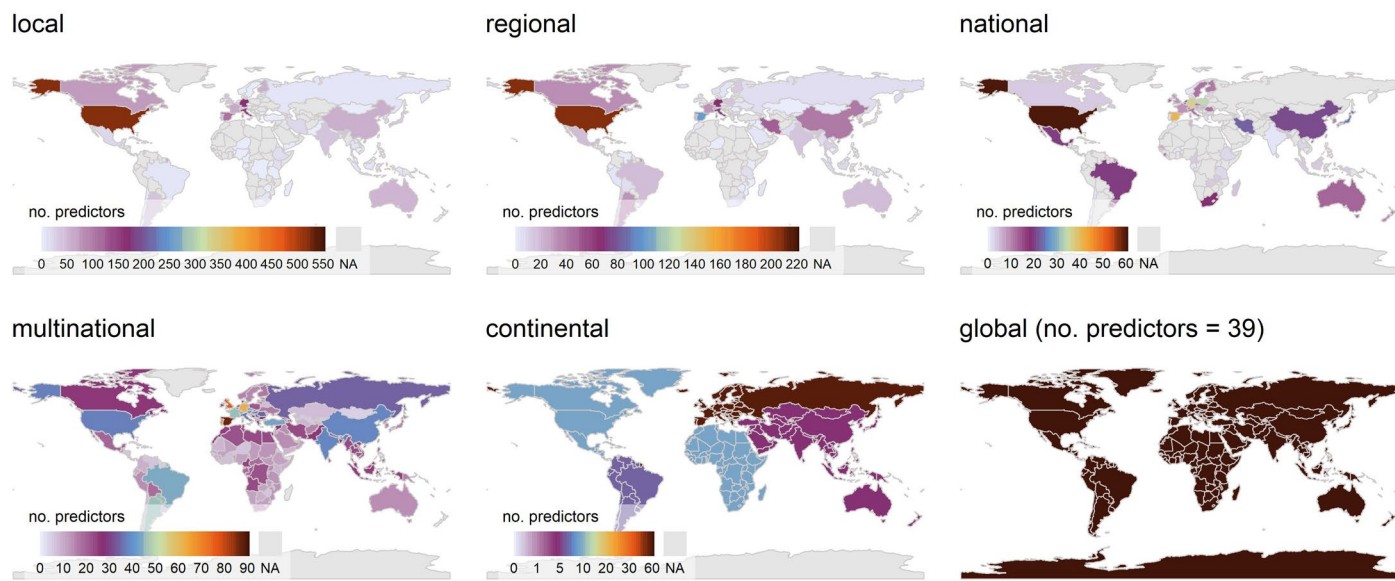

**Extended Data Fig. 8 | Spatial distribution of the total number of human predictors used in SDMs across various spatial scales.** These represent 2,307 human predictors used in 1,429 SDM articles published from 2000 to 2021. Note that marine articles are appended to their respective countries.

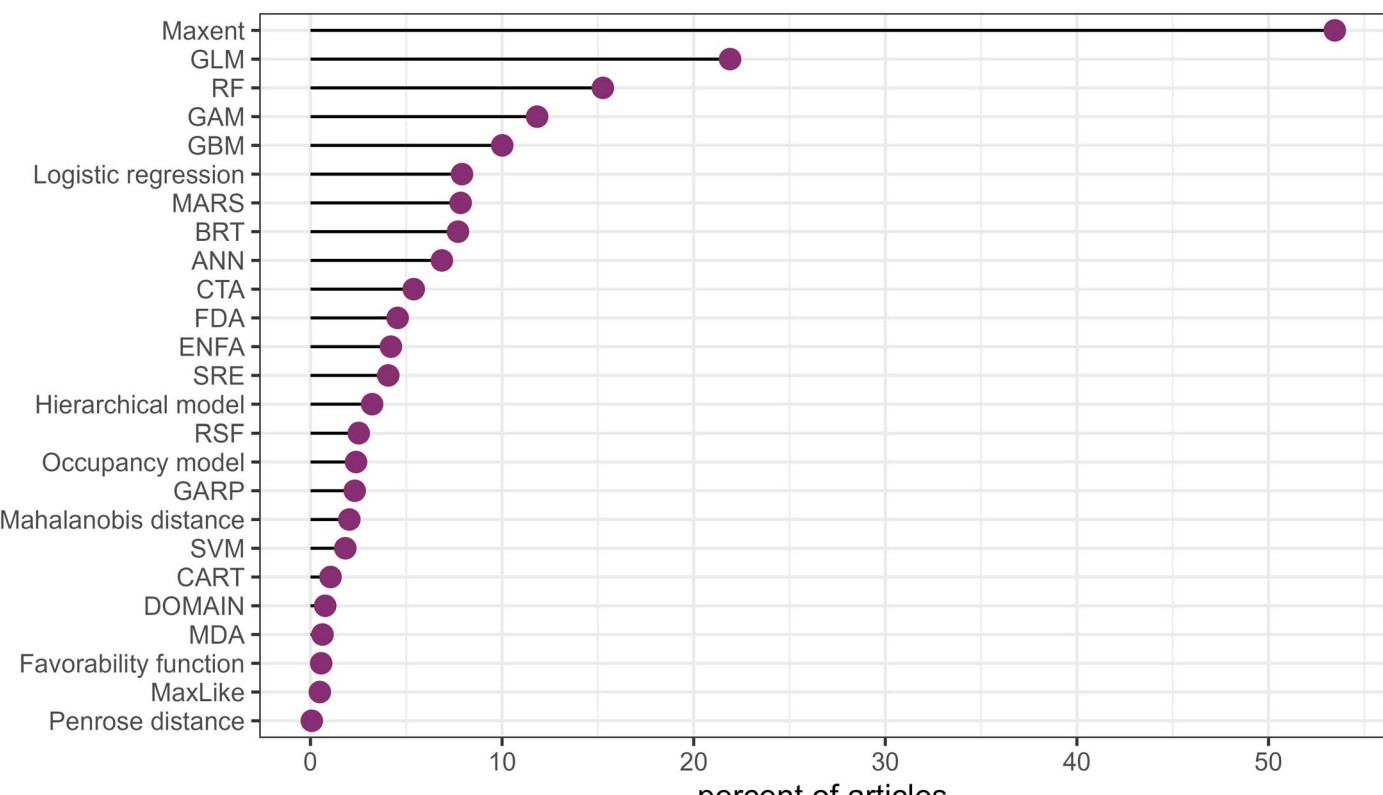

**Extended Data Fig. 9 | SDM algorithms used in modelling human influence on species distributions and the percent of articles that used them.** Most articles used Maxent, Generalized Linear Models, and Random Forest. For 299 of the 1,429 full articles (21%), multiple SDM algorithms were used, either separately or as an ensemble. *Abbreviations: ANN (artificial neural network); CTA (classification tree analysis, including classification and regression trees [CART]); Discriminant (discriminant analyses, including flexible and mixture [FDA; MDA]); DOMAIN (also known as Gower's distance); ENFA (environmental niche factor analysis); Favorability (favorability function); GAM (generalized additive model); GARP (genetic algorithm for rule-set production); GBM (gradient boosting model, including TreeNet and boosted regression trees [BRT]); GLM (general/generalized linear model, including logistic regression and resource selection function [RSF]); Hierarchical (a hierarchical model; typically a customized learning method such as Bayesian inference or occupancy model); MARS (multivariate adaptive regression splines); Mahalanobis (Mahalanobis distance, including Penrose distance); Maxent (maximum entropy); RF (random forest); SRE (surface range envelope, also known as BIOCLIM); SVM (support vector machine).

# Reporting Summary

## Statistics

For all statistical analyses, confirm that the following items are present in the figure legend, table legend, main text, or Methods section.

| n/a | Confirmed | |
|---|---|---|
| ☐ | ☒ | The exact sample size (*n*) for each experimental group/condition, given as a discrete number and unit of measurement |
| ☐ | ☒ | A statement on whether measurements were taken from distinct samples or whether the same sample was measured repeatedly |
| ☒ | ☐ | The statistical test(s) used AND whether they are one- or two-sided<br>*Only common tests should be described solely by name; describe more complex techniques in the Methods section.* |
| ☒ | ☐ | A description of all covariates tested |
| ☒ | ☐ | A description of any assumptions or corrections, such as tests of normality and adjustment for multiple comparisons |
| ☐ | ☒ | A full description of the statistical parameters including central tendency (e.g. means) or other basic estimates (e.g. regression coefficient) AND variation (e.g. standard deviation) or associated estimates of uncertainty (e.g. confidence intervals) |
| ☒ | ☐ | For null hypothesis testing, the test statistic (e.g. *F*, *t*, *r*) with confidence intervals, effect sizes, degrees of freedom and *P* value noted<br>*Give P values as exact values whenever suitable.* |
| ☒ | ☐ | For Bayesian analysis, information on the choice of priors and Markov chain Monte Carlo settings |
| ☒ | ☐ | For hierarchical and complex designs, identification of the appropriate level for tests and full reporting of outcomes |
| ☒ | ☐ | Estimates of effect sizes (e.g. Cohen's *d*, Pearson's *r*), indicating how they were calculated |

*Our web collection on statistics for biologists contains articles on many of the points above.*

## Software and code

Policy information about availability of computer code

| | |
|---|---|
| Data collection | The data for the articles collected from Web of Science are available on Figshare (https://doi.org/10.6084/m9.figshare.24225316). |
| Data analysis | All data were analyzed using R version 4.3.0. All code for this study are publicly accessible on GitHub (https://github.com/vffrans/Human_influence_SDMs) and Figshare (https://doi.org/10.6084/m9.figshare.24225316). |

For manuscripts utilizing custom algorithms or software that are central to the research but not yet described in published literature, software must be made available to editors and reviewers. We strongly encourage code deposition in a community repository (e.g. GitHub). See the Nature Portfolio guidelines for submitting code & software for further information.

## Data

Policy information about availability of data

All manuscripts must include a data availability statement. This statement should provide the following information, where applicable:
- Accession codes, unique identifiers, or web links for publicly available datasets
- A description of any restrictions on data availability
- For clinical datasets or third party data, please ensure that the statement adheres to our policy

The datasets developed from this study are in Supplementary Tables 5 and 6. They are also publicly accessible on the Figshare repository (https://doi.org/10.6084/m9.figshare.24225316).

## Human research participants

Policy information about studies involving human research participants and Sex and Gender in Research.

| | |
|---|---|
| Reporting on sex and gender | not collected |
| Population characteristics | *Describe the covariate-relevant population characteristics of the human research participants (e.g. age, genotypic information, past and current diagnosis and treatment categories). If you filled out the behavioural & social sciences study design questions and have nothing to add here, write "See above."* |
| Recruitment | *Describe how participants were recruited. Outline any potential self-selection bias or other biases that may be present and how these are likely to impact results.* |
| Ethics oversight | *Identify the organization(s) that approved the study protocol.* |

Note that full information on the approval of the study protocol must also be provided in the manuscript.

# Field-specific reporting

Please select the one below that is the best fit for your research. If you are not sure, read the appropriate sections before making your selection.

☐ Life sciences      ☐ Behavioural & social sciences      ☒ Ecological, evolutionary & environmental sciences

For a reference copy of the document with all sections, see [nature.com/documents/nr-reporting-summary-flat.pdf](http://nature.com/documents/nr-reporting-summary-flat.pdf)

# Ecological, evolutionary & environmental sciences study design

All studies must disclose on these points even when the disclosure is negative.

| | |
|---|---|
| Study description | This is a systematic review of articles on species distribution modeling (SDMs). We assessed whether articles published up to 2021 included human predictors in their models, and summarized those' articles procedures. |
| Research sample | Using Web of Science, we searched for all published articles up to December 31, 2021 that used the following terms in their titles, keywords, or abstracts: TS=(("species distribution model*" OR "environmental niche model*" OR "species niche model*" OR "bioclimatic niche model*" OR "habitat suitability model*" OR "ecological niche model*" OR "habitat model*")) AND DT=(Article) AND PY=(1900-2021)). This yielded 12,854 articles for the abstract screening step. |
| Sampling strategy | We screened all 12,854 article abstracts, searching for abstracts that indicate some acknowledgment of human influence on species' distributions. We manually screened ~300 abstracts at a time, added human-related terms found in those abstracts to a text-mining dictionary string, and then searched along the entire pool of abstracts to accept articles based on the updated terms. We repeated this for 28 iterations, allowing us to manually screen all rejected article abstracts (n=7,506), manually accept 551 article abstracts, automatically accept 4,626 article abstracts from the 477 terms added to the search, and manually review a total of 5,177 full articles and their supplementary materials (see PRISMA framework in Extended Data Fig. 1). |
| Data collection | All data were collected by the corresponding author, Veronica F. Frans. The 5,177 full articles that were accepted in the abstract screening step were manually downloaded based on a web search using each article's DOI or title. The supplementary materials of these articles were also collected. Fourteen accepted articles were not available. The full articles were then reviewed to see whether human predictors were used in species distribution models. This led to 1,429 eligible articles for our analysis and summary. We summarized the 1,429 articles by gathering the general focus (or, aim) of the study (as stated by the authors in the abstract or introduction), spatial scale of the study area, study area countries, the study's time frame (past, present and/or future SDM training and projection), the time frame represented by human predictors (including simulated scenarios across time), the taxa studied, study domain (terrestrial, marine, or freshwater habitat type), and SDM algorithms. For each article, we also listed the human predictors' names and the total numbers of environmental predictors used in the SDMs. We provide a description of these data in Supplementary Table 2. |
| Timing and spatial scale | We conducted the Web of Science search on September 14, 2022. The articles were collected from the year 1900 to 2021, but our analysis was restricted to articles published between 2000 and 2021. We did not limit the geographic coverage of our study. |
| Data exclusions | In the abstract screening step, we did not accept articles that did not mention human influence on species distributions in the abstract. Of the 5,177 full articles that we read, we did not accept articles that did not use human predictors within species distribution models. |
| Reproducibility | We maximized the reproducibility of our work by doing all abstract screening and data cleanup and analysis in R. The only portion of our work that cannot be automated is the collection of the raw data for each of the 1,429 accepted articles. These data were typed up in Excel while reading the articles. The PDFs of these articles were highlighted and annotated, and cannot be distributed due to copyright issues with article publications. However, all raw data corrections from the Excel spreadsheet were done in code in R. This project has 5 R scripts, which are accessible on GitHub (https://github.com/vffrans/Human_influence_SDMs) and Figshare (https://doi.org/10.6084/m9.figshare.24225316). |

| Randomization | We did not do randomization in our study. We were able to summarize all 1,429 accepted articles for our review and analysis. |
|---|---|
| Blinding | We did not do any blinding procedures in our study, since it was systematic review. |

Did the study involve field work? ☐ Yes ☒ No

# Reporting for specific materials, systems and methods

We require information from authors about some types of materials, experimental systems and methods used in many studies. Here, indicate whether each material, system or method listed is relevant to your study. If you are not sure if a list item applies to your research, read the appropriate section before selecting a response.

## Materials & experimental systems

| n/a | Involved in the study |
|---|---|
| ☒ ☐ | Antibodies |
| ☒ ☐ | Eukaryotic cell lines |
| ☒ ☐ | Palaeontology and archaeology |
| ☒ ☐ | Animals and other organisms |
| ☒ ☐ | Clinical data |
| ☒ ☐ | Dual use research of concern |

## Methods

| n/a | Involved in the study |
|---|---|
| ☒ ☐ | ChIP-seq |
| ☒ ☐ | Flow cytometry |
| ☒ ☐ | MRI-based neuroimaging |

