## [Peer Review File · Nature Ecology & Evolution]

Peer Review Information

Journal: Nature Ecology & Evolution

Manuscript Title: Gaps and opportunities in modeling human influence on species distributions in the Anthropocene

Corresponding author name(s): Veronica F. Frans

Editorial Notes:

Reviewer Comments & Decisions:

Decision Letter, initial version:

10th November 2023

Dear Ms Frans,

Your manuscript entitled "Gaps and opportunities in modeling human influence on species distributions in the Anthropocene" has now been seen by two reviewers, whose comments are attached. While Reviewer 1 is quite positive, Reviewer 2 has raised a number of concerns which will need to be addressed before we can offer publication in Nature Ecology & Evolution. We will therefore need to see your responses to the concerns, along with a revised manuscript, before we can reach a final decision regarding publication.

We therefore invite you to revise your manuscript taking into account all reviewer comments. Please highlight all changes in the manuscript text file.

- * Include a "Response to reviewers" document detailing, point-by-point, how you addressed each reviewer comment. If no action was taken to address a point, you must provide a compelling argument. This response will be sent back to the reviewers along with the revised manuscript.
- * If you have not done so already please begin to revise your manuscript so that it conforms to our Analysis format instructions at <http://www.nature.com/natecolevol/info/final-submission>. Refer also to any guidelines provided in this letter.
- * Include a revised version of any required reporting checklist. It will be available to referees (and, potentially, statisticians) to aid in their evaluation if the manuscript goes back for peer review. A revised checklist is essential for re-review of the paper.

2[REDACTED]

Nature Ecology & Evolution is committed to improving transparency in authorship. As part of our efforts in this direction, we are now requesting that all authors identified as 'corresponding author' on published papers create and link their Open Researcher and Contributor Identifier (ORCID) with their account on the Manuscript Tracking System (MTS), prior to acceptance. ORCID helps the scientific community achieve unambiguous attribution of all scholarly contributions. You can create and link your ORCID from the home page of the MTS by clicking on 'Modify my Springer Nature account'. For more information please visit please visit www.springernature.com/orcid.

[REDACTED]

Reviewer expertise:

Reviewer #1: SDMs, evidence synthesis

Reviewer #2: Ecological modeling, human influence, species movements

Reviewers' comments:

Reviewer #1 (Remarks to the Author):

The influence of human on species distributions is widely known and many human predictors have been used in species distribution models (SDMs). It is necessary to do a synthesis to get to know the current status and identify the gaps. The authors did a systematic review of the literature on this topic. They found 1439 papers used human predictors among the 12854 SDM papers, and these studies used 2354 unique human predictors. Interestingly, they proposed 18 questions, which will be useful for advancing the research in this field. Their suggestion to make a list of useful human

2predictors with data publicly accessible is even more interesting.

This paper is well-written, and the methodology used is sound. I only have a minor concern. For SDM algorithms used across studies (in Extended Data Fig. 8), some algorithms can be combined. These include GLM and Logistic regression, CART and CTA, GBM and BRT. Algorithms in each pair do the same work for SDM since the response variable is binary.

Canran Liu

Reviewer #2 (Remarks to the Author):

This literature review investigates the use of human influence variables in species distribution models (SDMs) across taxa and scales. An impressive effort – well documented with code – has been undertaken to extract SDM publications and their associated environmental predictor variables to derive a comprehensive picture about if and how human influence is included in modelling efforts. Main conclusions are the continuing lack of understanding regarding human influence impacts and the lack of a standardised variable set. Further findings are that human influence variables are often kept constant across time, while e.g. global climate models exist for future projections. The compiled database is impressive and the analysis worth being published; I also very much appreciate the theory-driven considerations, like the niche-concept or community assembly rules, to advance the understanding of human influence variables in SDMs.

Having said that, I am questioning a bit the novelty and originality of the research in respect to a journal like *Nature Ecology and Evolution*, as the whole study is a bit descriptive and main problems and flaws on missing out human influence variables on model results are in fact very general modelling issues. These are for example neglecting key variables, using pseudo-correlating variables or introducing biases by imbalanced model designs. These issues have been in length addressed in specific modelling literature and are not novel.

I also suggest some changes to better distil the message: Specifically, it appears that the timeline concerning the emergence of recent datasets and initiatives has not received sufficient attention. For instance, the Global Human Footprint map was only released in 2002 (Sanderson et al., 2002, *BioScience*), and similar timing applies to global initiatives like Movebank or GBIF, which became available in the early 2000s. Moreover, the field of urban wildlife ecology has recently gained significant momentum. Consequently, it might be misleading to generalize that human influence has not been adequately addressed in all literature since 1980 without considering these advancements in spatial data availability and research areas to be more recent. To provide a more comprehensive assessment, I propose segregating the analysis into two periods, namely pre- and post-2000, which would offer a clearer perspective and emphasize the significance of global initiatives.

Furthermore, I observed a lack of a distinct and rigorous definition for human influence in the study. This influence can manifest both directly and be quantifiably measured through environmental variables, as well as indirectly through effects that render seemingly suitable habitats unsuitable – a phenomenon we may term 'cryptic human influence.' It is important to introduce and analyse this concept separately. For instance, it should be identified which Species Distribution Models (SDMs) failed to account for unmeasured spatial effects, such as hunting pressure, in addition to the directly measured ones to learn from this huge analysis effort.

3In the context of implicit vs. explicit human influence variables: Another aspect that warrants more comprehensive exploration is the notion of ecological sinks or traps within SDMs. In other words, certain areas may exhibit high suitability based on a range of environmental and biotic predictors, but human influence renders these areas unsuitable. This dilemma necessitates a two-step approach, akin to the one introduced by Naves et al. in 2003 in their study on endangered brown bears in northern Spain ("Endangered species constrained by natural and human factors: the case of brown bears in northern Spain," *Conserv. Biol.* 17: 1276 – 1289), or the application of approaches that encompass population dynamics. Identifying situations where explicit variables are approximating implicit ones would strengthen this study.

Specific comments:

Line 44: 1,439 of how many SDM papers in total?

Line 64: land cover is listed here as abiotic variable; however, land cover is also a biotic variable; presence of forest might mean prey or mate availability or absence of human influence. This variable definitely needs to be considered as human influence variable. Or, a clear definition of human influence beyond measurable geographic variables is needed.

Lines 80-89: This is a general problem of an inappropriate model design and analysis and not specific to human influence variables in SDMs.

Line 110: I find this statement that modelling human influence is rarely done in SDMs quite disturbing here, as the reader does not yet know how human influence was defined, extracted and analysed. I.e. given that land use/ land cover is a standard variable in SDMs that can be a direct measure of human influence, I was quite puzzled about this finding. It means land cover is not a standard variable in SDMs?

Line 132: the trend that continental studies are few is maybe reflecting the fact that global initiatives and datasets are only recently available. These figures might be just a publication bias? Please crosscheck.

Fig 4A: The ~ 40 studies that form a cluster using ~100 variables seem to be outliers: I wonder to what extent this is driven by globally available standard geodatasets (like Bioclim aka WorldClim) and their derivatives (like slope, aspect, TRI from DEMs) and early MaxEnt modelling studies that blindly confronted their data with all available variables. Using few predictors for human influence does not mean models cannot well discriminate the training data. Models based on hypothesis testing and model selection generally use few variables. Again, these are general modelling philosophies unrelated to the inclusion of human influence.

Line 200: That human footprint was only used in 2% of the selected SDM publications might again be a bias of the literature study not accounting for the fact that this variable was only available > 2003, but the literature study pools all findings since 1980.

Lines 212 and following: The chapter on SDG assessment comes as a surprise here and maybe a bit distracting. What is the target of this chapter? An assessment of the SDG goals needs to be formulated more context specific, as the SDG goals are so broadly termed and would fit every modelling purpose.

Line 235. This is an important finding that forecasting studies kept human influence variables constant, but again a general problem of the modelling procedure, not only related to human influence. Maybe add some suggestions of how predictive models of human influence could be generated (e.g. in the case of logging and forest loss: Gaveau, D.L.A., et al. (2013). *Reconciling forest conservation and logging in Indonesian Borneo.* *PLoS ONE* 8, e69887.).

Fig 6. Sorry, I do not understand this figure. It is hard to discover an 'arrow' here, especially in the

4large blue area.

Lines 256 and following: Assessing SDM fit. This chapter is not really well developed and very descriptive, as the findings are maybe very context dependent. E.g. the models that were improved with human influence but not chosen as best models might be those for which no future projections were available. As a reader, I do not know what knowledge to gain from this chapter.

Lines 277 and following: I like the Box and the questions there; most of these questions should be posed before developing a model, actually, and can be a nice guidance before setting up models. It could be streamlined by taking out some of the bullet points that are overly general modelling issues (e.g. line 286, line 303, line 311, line 314)

Line 348. I suggest citing some seminal work by Lenore Fahrig on habitat fragmentation here

Lines 366: these ideas are similarly to the Essential Biodiversity Variable (EBV) discussions – please acknowledge this initiative here. (E.g., Jetz W, et al. Essential biodiversity variables for mapping and monitoring species populations. Nat Ecol Evol. 2019 Apr;3(4):539-551. doi: 10.1038/s41559-019-0826-1.)

Lines 455-459: these are general modelling flaws when predicting SDMs beyond data boundaries. I suggest deleting this paragraph

Line 479: what about search terms such as occupancy model or resource selection models or niche modelling?

Extended Data Table 1: How/ where was logging included as the main human disturbance source responsible for biodiversity declines in tropical regions?

*****END*****

Author Rebuttal to Initial comments

Response to Reviewers

(Reviewers' Comments in bold font and Authors' Responses in regular font)

NATECOLEVOL-23102331

“Gaps and opportunities in modeling human influence on species distributions in the Anthropocene”

Reviewer expertise:

Reviewer #1: SDMs, evidence synthesis

Reviewer #2: Ecological modeling, human influence, species movements

A. Reviewer #1

Reviewer #1 (Remarks to the Author):

The influence of human on species distributions is widely known and many human predictors have been used in species distribution models (SDMs). It is necessary to do a synthesis to get to know the current status and identify the gaps. The authors did a systematic review of the literature on this topic. They found 1439 papers used human predictors among the 12854 SDM papers, and these studies used 2354 unique human predictors. Interestingly, they proposed 18 questions, which will be useful for advancing the research in this field. Their suggestion to make a list of useful human predictors with data publicly accessible is even more interesting.

This paper is well-written, and the methodology used is sound. I only have a minor concern. For SDM algorithms used across studies (in Extended Data Fig. 8), some algorithms can be combined. These include GLM and Logistic regression, CART and CTA, GBM and BRT. Algorithms in each pair do the same work for SDM since the response variable is binary.

Canran Liu

Dear Dr. Liu,

Thank you so much for your evaluation of our manuscript and your interest in the findings of our work! We also thank you for sharing this minor concern about Extended Data Fig. 8 (now Extended Data Fig. 9 in the revised manuscript). We followed your advice and condensed our listed algorithms, and the figure looks much better now. Thank you!

Please also note that we have addressed concerns from Reviewer #2 that further improved our manuscript. We hope you will find this new, updated manuscript to be just as impressive.

Thank you for your time!

Sincerely,

Veronica Frans and Jianguo (Jack) Liu

B. Reviewer #2

Reviewer #2 (Remarks to the Author):

6This literature review investigates the use of human influence variables in species distribution models (SDMs) across taxa and scales. An impressive effort – well documented with code – has been undertaken to extract SDM publications and their associated environmental predictor variables to derive a comprehensive picture about if and how human influence is included in modelling efforts. Main conclusions are the continuing lack of understanding regarding human influence impacts and the lack of a standardised variable set. Further findings are that human influence variables are often kept constant across time, while e.g. global climate models exist for future projections. The compiled database is impressive and the analysis worth being published; I also very much appreciate the theory-driven considerations, like the niche-concept or community assembly rules, to advance the understanding of human influence variables in SDMs.

Having said that, I am questioning a bit the novelty and originality of the research in respect to a journal like Nature Ecology and Evolution, as the whole study is a bit descriptive and main problems and flaws on missing out human influence variables on model results are in fact very general modelling issues. These are for example neglecting key variables, using pseudo-correlating variables or introducing biases by imbalanced model designs. These issues have been in length addressed in specific modelling literature and are not novel.

Dear Reviewer #2,

Thank you so much for your time in reading our manuscript and reviewing our additional materials!

We are so excited that you found our work interesting and worth publication, and we are very grateful to you for all the help that you have provided to better streamline the message of our work and clarify the distinct contributions of our manuscript to the field and to this journal. We also appreciate the extra time that you have put in to provide us with an additional list of papers to review so we can properly address your concerns regarding novelty and originality. You will see that we have addressed all your concerns, using this opportunity to improve the impact and relevance of our research.

Mainly, we have made the following changes to the manuscript in response to your review:

1. We provide a definition for human predictors at the start of our results section.
2. We changed the years of the study to only comprise articles after the year 2000, as to remove potential misunderstandings about human predictor availability versus human predictor utility; this removes 9 articles from our analysis, so there were minimal changes to the results and message of our study.
3. We created new figures on data use and added them to Fig. 2C, Fig. 4, and Extended Data Fig. 7 to help address concerns about publication bias and data availability around the world or over time.

74. We revised Figure 6 for easier interpretation, with a more detailed legend and guidelines on how to interpret the figure in the captions, as well.
5. We deleted a paragraph related to general issues found in SDMs.
6. We removed four questions from Box 1 that could be interpreted as general species distribution modeling questions.
7. Regarding the novelty and originality of our analysis, we received your reply from Dr. McKay when we asked for the references from which your concerns had stemmed (thank you!). We read the articles and highlight the differences between our work and theirs at the end of this document. Because we also found these additional references to be quite valuable, we also cited them in the Discussion section to direct future research.

Please see our specific responses below, and we look forward to your feedback.

We are very excited about the changes we made thanks to your insights, and we hope you are also excited about them, too.

Thank you again for your time and consideration!

Sincerely,

Veronica Frans and Jianguo (Jack) Liu

1. Concern #1

I also suggest some changes to better distil the message: Specifically, it appears that the timeline concerning the emergence of recent datasets and initiatives has not received sufficient attention. For instance, the Global Human Footprint map was only released in 2002 (Sanderson et al., 2002, BioScience), and similar timing applies to global initiatives like Movebank or GBIF, which became available in the early 2000s. Moreover, the field of urban wildlife ecology has recently gained significant momentum. Consequently, it might be misleading to generalize that human influence has not been adequately addressed in all literature since 1980 without considering these advancements in spatial data availability and research areas to be more recent. To provide a more comprehensive assessment, I propose segregating the analysis into two periods, namely pre- and post-2000, which would offer a clearer perspective and emphasize the significance of global initiatives.

Thank you for sharing that major data trends could have the potential to influence the use of human predictors in SDMs. As you have stated, global initiatives such as Movebank and GBIF may play a role in the spike of SDM literature over the years. As shown in the original Figure 1 of our manuscript, which we

8provide below, we can indeed see a spike in the number of general SDM articles after the year 2000. Thus, there is an exponential growth in the number of SDM articles over time. However, when focusing on the proportion (or percent) of SDM articles per year that use human predictors within their models, our analysis shows that despite the spike in SDM articles, there has been a plateau in the relative interest in modeling human influence on species distributions (<15% of published SDM articles per year) since the early 2000s. Thus, while global initiatives have been made and data are becoming more and more available, there is not a clear uptick in the use of human predictors in SDMs in response to these data initiatives.

To avoid misinterpretation of our article's message, and also because the number of SDM articles prior to the year 2000 were pretty small, we cut off the earlier years of our synthesis. Our analysis now starts from the year 2000 and ends in the year 2021. This cuts out 183 total articles from the original 12,854 articles (1.42% reduction), 76 articles from the 5,177 full articles that we read that acknowledged human influence in the abstracts (1.46% reduction), and 9 articles from 1985 to 1999 that modeled human influence on species distributions that were part of our analysis (0.63% reduction in articles from our original dataset of 1,439 articles). This is a minor percentage of articles, and the changes to our results and final dataset are also minor, as you will see in some of our other responses and tracked changes throughout the manuscript.

We thus have made the following changes in response to your suggestions:

1. We limited the scope of our analysis to the years 2000 to 2021 (L537-538).
2. We added two additional data fields to Dataset 2 showing the first and last year that each human predictor was used in a publication (also in Supporting Information Table S4). We also illustrated this in a new figure, Fig. 4 (see Concern #8).
3. From these years of use, we made an additional figure (Fig 2C), showing the temporal spread of human predictor use across various spatial scales.

2. Concern #2

Furthermore, I observed a lack of a distinct and rigorous definition for human influence in the study. This influence can manifest both directly and be quantifiably measured through environmental variables, as well as indirectly through effects that render seemingly suitable habitats unsuitable – a phenomenon we may term 'cryptic human influence.' It is important to introduce and analyse this concept separately. For instance, it should be identified which Species Distribution Models (SDMs) failed to account for unmeasured spatial effects, such as hunting pressure, in addition to the directly measured ones to learn from this huge analysis effort.

Human influence and human predictors refer to “human activities, presence, or pressures” (L76; L112). We again define human predictors in L524-525 of the Methods as follows: “Human predictors, also known as anthropogenic predictors, are those that include an indicator of human activities, presence, or pressures. These include predictors that directly allude to human influence (e.g., *human population size, human footprint, distance from residential areas*) or indirectly allude to human influence (e.g., *protected versus unprotected areas, land use/land cover*).”

In terms of “cryptic human influence,” we believe that you are describing what we refer to as “ambiguous” human predictors. We define “ambiguous predictors” as predictors that “can either represent human influence or be equally interpreted as environmental predictors.” (L217-218; also in L190-191). In our work, we have catalogued 490 articles that collectively use a total of 115 ambiguous predictors (L219).

For predictors such as “hunting pressure,” we have assigned a data type of “index,” but do not identify them as “unmeasured spatial effects.” While our analysis is expansive, we are also inspired by the many other ways that our synthesized list of human predictors can be expanded for more inquiries. We demonstrate such an expansion when we text-mined through the predictor list to test their relevance to the various Sustainable Development Goals. However, due to the heterogeneity of SDM studies (various taxa, focuses of study, and 1,936 human predictors being used only once across studies), it would be difficult to catalogue what the authors are missing as opposed to what the authors have chosen to do in their modeling procedures. Nevertheless, we are confident that our work will spark more interest in the

10subject of human influence on species distributions, where new SDM frameworks and guidelines can be developed.

3. Concern #3

In the context of implicit vs. explicit human influence variables: Another aspect that warrants more comprehensive exploration is the notion of ecological sinks or traps within SDMs. In other words, certain areas may exhibit high suitability based on a range of environmental and biotic predictors, but human influence renders these areas unsuitable. This dilemma necessitates a two-step approach, akin to the one introduced by Naves et al. in 2003 in their study on endangered brown bears in northern Spain ("Endangered species constrained by natural and human factors: the case of brown bears in northern Spain," *Conserv. Biol.* 17: 1276 – 1289), or the application of approaches that encompass population dynamics. Identifying situations where explicit variables are approximating implicit ones would strengthen this study.

Thank you for mentioning this topic and providing this article, which was also one that we found during our analysis. We agree that unsuitability after including human predictors in SDMs is a topic of concern. We originally alluded to it in L72-74 and L79-89 of our Introduction. Thanks to your suggestion, we added it to Box 1C as follows:

“Which human predictors are the most helpful for identifying ecological sinks or traps?”

We also highlighted this idea in our Discussion for future applications (L462-463):

“Evaluating SDM projections with and without human predictors can also assist in identifying and mapping ecological traps or sinks for critical species¹⁰¹.”

4. Concern #4

Specific comments:

Line 44: 1,439 of how many SDM papers in total?

We changed the text to the following: “From a search of 12,854 articles, we found only 1,429 articles using human predictors within SDMs.”

5. Concern #5

Line 64: land cover is listed here as abiotic variable; however, land cover is also a biotic variable; presence of forest might mean prey or mate availability or absence of human influence. This variable

definitely needs to be considered as human influence variable. Or, a clear definition of human influence beyond measurable geographic variables is needed.

Thank you for catching that. Yes, land cover can also include biotic interactions and it can be a human predictor. We deleted land cover as an example here. We do classify land cover as a human predictor in the rest of our manuscript, and it is also in our dataset.

6. Concern #6

Lines 80-89: This is a general problem of an inappropriate model design and analysis and not specific to human influence variables in SDMs.

Yes, this problem can happen when using environmental predictors, as well. While this problem is not *exclusive* to the use of human predictors, it is a common issue which we found among the papers that model human influence on species distributions. It also has great potential in affecting conservation, policy, decision-making, and other applications. We thus wanted to allude to this issue in the introduction, while later in the results we provide more details and context (e.g., L417-433; 453-487). We changed the text to focus on the issue in the context of broader, real-world applications of SDMs, with the following text changed in L86-89:

“Thus, inadequately accounting for human predictors in species projections could largely affect broader applications or interpretations from SDMs²³, leading to false optimism about a species’ future trajectory or the implementation of misinformed policies.”

7. Concern #7

Line 110: I find this statement that modelling human influence is rarely done in SDMs quite disturbing here, as the reader does not yet know how human influence was defined, extracted and analysed. I.e. given that land use/ land cover is a standard variable in SDMs that can be a direct measure of human influence, I was quite puzzled about this finding. It means land cover is not a standard variable in SDMs?

Because the writing format for research articles in Nature Ecology and Evolution is Introduction, Results, Discussion, and lastly, the Methods, we tried our best to bring the readers straight to the results while still offering some context without repeating the Methods in the Results section. In the Introduction, we define “human predictors” as “predictors relating to human activities, presence, or pressures” (L76-77). We added this definition to L112 as a refresher for the reader as they begin reading the results. Thank you for informing us of the confusion.

The use of human predictors within an SDM indicates that human influence on species distributions is being modeled by the author. In our methods, we classify land use/land cover as a human predictor. It is one of the most frequently used human predictors, used by 17% of the summarized articles. While it is used at the highest percentage, it is not a standard predictor for modeling human influence on species' distributions.

8. Concern #8

Line 132: the trend that continental studies are few is maybe reflecting the fact that global initiatives and datasets are only recently available. These figures might be just a publication bias? Please crosscheck.

Thank you for this idea! We removed all articles prior to the year 2000 from our analysis to avoid misinterpretation about data availability in relation to data initiatives, and also because the articles were few (9 out of the original 1,439 papers of the study). We also created additional figures which we added to Fig. 2C, Fig. 4, and Extended Data Fig. 7 (please note that we moved the original Extended Data Table 2 to Supporting Information in order to fit our new figures within the main manuscript). These figures show (1) the first published years of human predictor use in SDMs, mapped globally across multiple spatial scales of study (Fig. 2C); (2) the number of unique human predictors used in SDMs, mapped globally across multiple spatial scales (Extended Data Fig. 7); and (3) bubble scatterplots comparing the first and last (most recent) years that each of the 2,307 human predictors was used in an SDM, separated by 12 categories (Fig. 4). We thus state the following in L141-144:

“In such areas, it was not until around 2010 that human predictors were first used in SDMs at global and continental scales. In Africa, South America, and some parts of Asia especially, it was not until 2020 that human predictors were first used in SDMs at national, regional, or even local scales (Fig. 2C).”

We also say the following in L220-223:

“New human predictors have been consistently emerging each year (Fig. 4). The categories with the most momentum and persistence in use after first being introduced by authors or made available related to food and agriculture (n=125), infrastructure (n=85), and transportation (n=48).”

Finally, we describe this new analysis in L599-604 of the Methods.

9. Concern #9

Fig 4A: The ~ 40 studies that form a cluster using ~100 variables seem to be outliers: I wonder to what extent this is driven by globally available standard geodatasets (like Bioclim aka WorldClim) and their

derivates (like slope, aspect, TRI from DEMs) and early MaxEnt modelling studies that blindly confronted their data with all available variables. Using few predictors for human influence does not mean models cannot well discriminate the training data. Models based on hypothesis testing and model selection generally use few variables. Again, these are general modelling philosophies unrelated to the inclusion of human influence.

Thank you for this note. There seems to be a misunderstanding about the number of papers using over 100 predictors. We apologize if this figure has caused misinterpretation at first glance, but the light green color represents values less than 5. Looking further into them, they represent only two studies that use over 100 predictors in their SDMs, and not 40.

In Figure 4A and also in Dataset 1, the two studies using over 100 predictors are paper IDs (UID) 3951 and 910, respectively published in 2019 and 2021 as Kanagaraj et al. 2019 (Diversity and Distributions) and Conley et al. 2021 (Canadian Journal of Fisheries and Aquatic Sciences). Kanagaraj et al. used ensemble SDMs (GLM; GBM; GAM; ANN; SRE; CTA; RF; MARS; FDA; Maxent) and Conley used Random Forest. Some of the predictors used by Kanagaraj et al. were from Worldclim while Conley et al. did not use Worldclim. Hence, these articles are not examples of early Maxent modeling studies, as you have proposed.

We were also curious about the use of standardized datasets such as Worldclim and appreciate that you have considered them, as well. In Supplementary Dataset 1, we have a column that indicates whether an article uses Worldclim data in the SDM. 470 out of the 1,429 articles using human predictors in SDMs had also used Worldclim data in their models (33% of articles, mentioned in L389-396). As many of your other questions have shown, Supplementary Datasets 1 and 2 will be beneficial for readers wanting to explore the data and get at the root of some of these and other potential outliers of interest.

10. Concern #10

Line 200: That human footprint was only used in 2% of the selected SDM publications might again be a bias of the literature study not accounting for the fact that this variable was only available > 2003, but the literature study pools all findings since 1980.

As we noted in response to your comment above (Concern #1), we changed the scope of our study to the years 2000 to 2021 to avoid misinterpretation about the relationship between the start of data initiatives and the overall status of using human predictors in SDMs. We also noticed a spelling error in Dataset 2 that caused us to mistakenly calculate a lower number of articles for the Human Footprint. These changes brought the number of articles using Human Footprint to 74 (5.17%). We made edits to Dataset 2, Supporting Information Table S4, and L211.

11. Concern #11

Lines 212 and following: The chapter on SDG assessment comes as a surprise here and maybe a bit distracting. What is the target of this chapter? An assessment of the SDG goals needs to be formulated more context specific, as the SDG goals are so broadly termed and would fit every modelling purpose.

Apologies for that surprise. After looking back at our Introduction, we realized that we forgot to mention this analysis in the paragraph where we summarized what we did (L91-102). Thank you so much for catching that!

We added the following sentence to our introduction (L98-100):

“Acknowledging the critical intersection between biodiversity and sustainability^{19,25–27}, we also examined how these human predictors related to global Sustainable Development Goals²⁰.”

We also expanded our reasoning for this analysis in L237-240:

“As both global biodiversity conservation initiatives and Sustainable Development Goals (United Nations SDGs) are set for multiple targets by the years 2030 and 2050^{20,41}, trade-offs and synergies between species and human prosperity are inevitable²⁵. We thus tested whether the human predictors used for modeling species distributions related to any of the 17 SDGs.”

SDMs are used for various purposes quite broadly—from public health, to monitoring illegal activities, to conservation and policy, among others. We thus wanted to provide additional, objective ways to evaluate and subset the human predictors from our synthesis beyond the 12 categories and 6 data types as shown in Figure 4. This section on SDGs also contributes to our Discussion section on the broader applications of human predictors in SDMs (L471-481), where we state the following:

“While SDG indicators directly relating to species distributions have already been identified under SDG-14 (Life below Water) and SDG-15 (Life on Land), studies are continually emerging that show that species within protected areas are linked to other SDGs, like Decent Work and Economic Growth (SDG-8; tourism increasing the income around protected areas), Industry, Innovation, and Infrastructure (SDG-9; building roads around protected areas for access), and even Partnerships for the Goals (SDG-17; international conservation breeding programs introducing individuals to new locations)²⁷. Beyond protected areas, even human predictors pertaining to Peace, Justice and Strong Institutions (SDG-16) could correlate with species distributions, as issues such as systemic racism in urban areas can impact biodiversity at national scales¹⁰⁵. An assessment of species distribution changes over time in relation to the UN’s 231 SDG indicators and across multiple taxa may reveal the relevance of species to all sectors of global policy and human flourishing.”

Additionally, many ecologists are interested in coupled human and natural systems (CHANS) and SDGs (e.g., Blicharska et al. 2019, Nature Sustainability, DOI: 10.1038/s41893-019-0417-9; Clémençon 2021, Global Sustainability, DOI: 10.1017/sus.2021.14; Cooper et al. 2023 & Lubchenco et al. 2023, Nature Ecology and Evolution, DOI: 10.1038/s41559-023-02209-3 & 10.1038/s41559-023-02208-4; Rosin et al. 2019, Journal of Applied Ecology, DOI: 10.1111/1365-2664.13566). Besides the many publications by ecologists on the topics of SDGs, there have also been important sessions hosted at conferences addressing them (e.g., the Ecological Society of America in 2013 and 2017; <https://eco.confex.com/eco/2013/webprogram/Session8773.html>; <https://eco.confex.com/eco/2017/webprogram/Session13121.html>). Our work contributes to such emerging conversations on the intersection of species' geographic distributions and sustainable development.

Regarding the means by which we linked the SDGs to human predictors, while the SDGs themselves are broadly termed, their respective 231 unique indicators and 169 targets contribute to how the text-mining was informed. The `text2sdg` package in R uses a combination of 6 SDG query systems developed by experts such as Elsevier, the Sustainable Development Solutions Network (SDSN) Australia, New Zealand & Pacific Network, Aurora Universities Network, and others. The detection function combines these query systems with a trained ensemble model from expert evaluations of SDGs to assign SDGs to the list of human predictors. We thus find this labeling system to be sound.

In terms of modeling purposes, for each reviewed article, we recorded the focus of each study, based on statements from the authors in the abstracts and/or introduction section(s). Matches between modeling purpose and SDGs can be explored by researchers who are interested. We provide the study focus information in Dataset 1 and the predictor list in Dataset 2. An analysis comparing the modeling purpose of an article with the general human predictor categories or their corresponding SDGs is possible. We are thus excited by your comments, as these indicate further ways that the datasets from our study can be useful to the ecological community.

12. Concern #12

Line 235. This is an important finding that forecasting studies kept human influence variables constant, but again a general problem of the modelling procedure, not only related to human influence. Maybe add some suggestions of how predictive models of human influence could be generated (e.g. in the case of logging and forest loss: Gaveau, D.L.A., et al. (2013). Reconciling forest conservation and logging in Indonesian Borneo. PLoS ONE 8, e69887.).

Thank you for this example. We have discussed the issue and cited your reference in L447-450 of the Discussion alongside other suggestions:

16“One solution could be to simulate multiple potential percent increases or decreases of a predictor’s values or area coverage over time^{96,97} or to use propensity score matching⁹⁸ if mechanistic predictors of human influence are unavailable. Open-access tools to simulate land use change are also being developed⁹⁹.”

13. Concern #13

Fig 6. Sorry, I do not understand this figure. It is hard to discover an ‘arrow’ here, especially in the large blue area.

Thank you for the opportunity to carefully reconsider this figure and get more creative. We like our new figure a lot better now! We hope that you like it as well. The arrows are much easier to see because we changed the scale of the visualization. We also included a legend that breaks down how to interpret the “behavior” of the arrows. Finally, we edited the captions to include more information on how to interpret the figure (L279-285). Thanks again!

14. Concern #14

Lines 256 and following: Assessing SDM fit. This chapter is not really well developed and very descriptive, as the findings are maybe very context dependent. E.g. the models that were improved with human influence but not chosen as best models might be those for which no future projections were available. As a reader, I do not know what knowledge to gain from this chapter.

We edited L288-290 to clarify the knowledge to gain from this section: there is “no real ‘rule of thumb’ for human predictor selection and evaluation.”

Regarding your concerns for the development of this section due to potential context dependencies, we were aware of context dependencies (such as an author’s evaluation of the realism of model outcomes when human predictors are used), and noted them for 26 out of the 127 articles that compared environmental-only and environmental + human predictors in SDMs. Among these 26 articles, however, none of the authors stated that they chose environmental-only predictors due to a lack of future human predictor data, contrary to your suggestion. Instead, some authors chose to use human predictors because they improved predictions for future projections (e.g., Heubes et al. 2011, *Journal of Biogeography*). In cases where future projection data were not available, it was more common for authors to leave human predictors constant (L263-264; 122 articles). Additionally, there were 267 cases from the 5,177 full articles that we read in the full article screening step where human predictors were used outside of SDM training and projections as masks or additional intersections with SDM outputs (L117-118). Regarding the descriptive nature of this section, we summarize the results in Table S3 and refer to this table in L292.

17We believe that this section of the manuscript is important for future readers because it highlights the necessity for more detailed, constructive, model comparison research on using human predictors in SDMs, as we describe in Box 1B and L376-450. As you have indicated above when describing “implicit” and “explicit” predictor cases (Concern #3), it is possible that using human predictors in SDMs can tell us a different story about habitat suitability. Thus, the best way to understand such cases is also to evaluate assessments where environmental-only and human + environmental predictors are compared in SDM studies.

15. Concern #15

Lines 277 and following: I like the Box and the questions there; most of these questions should be posed before developing a model, actually, and can be a nice guidance before setting up models. It could be streamlined by taking out some of the bullet points that are overly general modelling issues (e.g. line 286, line 303, line 311, line 314)

Thank you so much! Yes, we want to make sure that this box is interesting and intriguing for readers and for future research. We deleted all the lines you have listed, and we agree that it sounds a lot better this way. Thank you for helping us to streamline our ideas.

16. Concern #16

Line 348. I suggest citing some seminal work by Lenore Fahrig on habitat fragmentation here

Thank you for this suggestion. We added the following citations:

Fahrig, L. & Rytwinski, T. Effects of roads on animal abundance: an empirical review and synthesis. *Ecology and Society* **14** (2009).

Fahrig, L. Effects of habitat fragmentation on biodiversity. *Annu. Rev. Ecol. Evol. Syst.* **34**, 487–515 (2003).

17. Concern #17

Lines 366: these ideas are similarly to the Essential Biodiversity Variable (EBV) discussions – please acknowledge this initiative here. (E.g., Jetz W, et al. Essential biodiversity variables for mapping and monitoring species populations. *Nat Ecol Evol.* 2019 Apr;3(4):539-551. doi: 10.1038/s41559-019-0826-1.)

Thank you for this citation. We appreciate the ideas of Jetz et al. 2019 and have mentioned it as a way for human predictors to be included in such a project. We added the following to L407-408 (the end of the paragraph you mentioned):

“Additionally, open data efforts such as the “Essential Biodiversity Variables” initiative⁸⁸ could include human predictors in their considerations.”

18. Concern #18

Lines 455-459: these are general modelling flaws when predicting SDMs beyond data boundaries. I suggest deleting this paragraph

Thank you for this suggestion. We deleted these lines from the manuscript.

19. Concern #19

Line 479: what about search terms such as occupancy model or resource selection models or niche modelling?

The search terms we have chosen are based on general synonyms and descriptions for species distribution modeling which have been provided by Franklin 2010—a long-standing resource on SDMs. The terms "occupancy models" and "resource selection models" (or "functions") are specific kinds of SDMs and are not general descriptions for SDMs. Using such terms would bias our search towards these specific ways of modeling species distributions. Additionally, "niche modelling" is more general than SDMs, which would lead to more false positives in our Web of Science search; instead, we use "species niche model*", "environmental niche model*", and "bioclimatic niche model*". To allow for additional variations in SDM descriptions, we use wildcards (*) in our search terms, where "model*", for example, would capture literature using "model", "models", "modeling", and "modelling".

Nevertheless, we thought it important to consider your suggested search terms. We searched Web of Science and began repeating our abstract screening, full article screening, and full article data extraction protocols and found that the results and message of our study would not change with an expanded search of articles. We had taken this suggestion seriously and contacted Senior Editor, Dr. McKay about your concerns. In response, Dr. McKay suggested that mentioning potential limitations due to search terms would be a reasonable approach. Thus, we emphasized the generality of terms in L485 (“...using search terms that were general and synonymous to SDMs...”) and added the following to our revised manuscript (L504-508):

“While we acknowledge that more articles could have been captured using additional search terms (e.g., listing SDM algorithms), a test using terms such as “occupancy model”, “resource selection function*”, or “niche model*” showed that our choice of general search terms and their resulting articles were sufficient to capture the current state of modeling human influence on species distributions.”

Thank you again for raising this question. We learned a lot from it, as it was helpful in building the confidence that our work is sufficient for our main goal of getting an overview on the state of modeling human influence on species distributions.

20. Concern #20

Extended Data Table 1: How/ where was logging included as the main human disturbance source responsible for biodiversity declines in tropical regions?

Thank you for this question. In Table 1, we describe predictors relating to disturbance as the following: “predictors describing habitat fragmentation, deforestation, degradation, change in naturalness, or indices of disturbance or avoidance.” This predictor category includes predictors such as logging. We added to this table some example names of predictors relating to logging. We also listed logging as an example of disturbance in L216 of the Results.

The dataset from our analysis is a benefit for readers and other researchers as they investigate additional questions on modeling human influence on species distributions. Your question is an indicator of the kinds of information that will be explorable once our dataset is made publicly available upon publication. In Supporting Information Table S3, as well as Supplementary Dataset 2, you will find 46 varieties of predictors related to logging (e.g., clear-cut areas, harvested forest, logging roads, logging sawmills, logging frequency). In Dataset 2, we have an extended list of the predictor table where such predictors are listed, and they have a list of the papers that use these predictors, where their unique paper ID (the “UID” column) corresponds to the sources listed in Dataset 1.

C. Additional Notes from Reviewer #2 via Email from Senior Editor, Dr. McKay

Additional notes from Reviewer #2 after an additional inquiry was emailed from the Editor on our behalf, to help address concerns about novelty and originality of our analysis:

201. Novelty Concern #1

E.g., regarding thoughts on which niche is represented:

-Araújo, M.B. and Guisan, A. (2006), **Five (or so) challenges for species distribution modelling**. *Journal of Biogeography*, **33**: 1677-1688. <https://doi.org/10.1111/j.1365-2699.2006.01584.x>

Thank you for this reference. We acknowledge that there is a long-standing conversation about the niche concept in species distribution modeling, for which we have cited numerous articles dating from as far back as 2000 to as recent as 2021 (L347-365). One recent, comprehensive work that has summarized the niche concept in SDMs is from Sales et al. 2021 (*Acta Oecologica*), where they conclude that SDMs model their own special kind of niche. There is also a series about whether to “ditch” “pitch” or “stitch” the niche concept, in a special issue in the *Journal of Biogeography* (McInerney & Etienne 2012; and Soberon 2014). Our analysis is novel because it adds an additional dimension and emphasis to the niche conversation for SDMs: we specifically focus on the niche concept when it intersects with human influence.

Araújo and Guisan 2006 talk about the Hutchinsonian framework with regard to biotic interactions (resources, constraints, and the realized and fundamental niche), but they do not discuss what happens to the niche concept in SDMs when human predictors are included in these models. In our manuscript, we specifically raise the question of the niche concept when human predictors are included in SDMs.

Later in their article, Araújo and Guisan mention that it is important to select meaningful predictors such as human disturbances, and we like this quote: “Nonetheless, it is reasonable to ask what else is left, when all the climate-related variance has been explained. Answering this question requires quantifying how much climate can explain species distributions compared to other predictors, such as soils, site history, human influences, or other factors.” It goes well with the list of articles that we reference in L348 at the beginning of the *Advancing ecological theory* section, so we cite the article there.

We are happy to see that back in 2006 Araújo and Guisan anticipated that there would be issues in predictor selection when combining human disturbances with commonly used predictors such as climate. However, with their article dating back to 2006, our search through over 12,800 SDM articles to date shows that distinct, directed efforts to interpret the variance explained by human predictors when combined with environmental predictors is still needed. We also liked the questions you raised about what you referred to as “cryptic human influence;” this also relates to Araújo and Guisan. In our work, we found only 127 articles that tested and compared SDM performance with and without human predictors (L287-300), but more work is needed. Our analysis thus serves as a way to call attention to the issues that persist beyond the issues raised by Araujo and Guisan back in 2006.

2. Novelty Concern #2

e.g., regarding transferability of models:

-Petitpierre, B., Broennimann, O., Kueffer, C., Daehler, C. and Guisan, A. (2017), Selecting predictors to maximize the transferability of species distribution models: lessons from cross-continental plant invasions. *Global Ecol. Biogeogr.*, 26: 275-287. <https://doi.org/10.1111/geb.12530>

Thank you for this reference. However, this paper examines the transferability of predictors using only climatic predictors. Yet, we agree that general lessons can be learned from it. For example, we admire this conclusion from the authors about predictor selection:

“For a majority of species, and from a purely predictive perspective, the best model is found using an iterative random approach (i.e. no strategy) to select the predictor dataset. Therefore, the variable selection providing the best model is species specific, meaning that the final combination of predictors should be carefully chosen based on its performance to explain the distribution of each individual species on independent data.”

Such statements emphasize the importance of our work. It is possible that the existence of over 2,300 human predictors (and counting) is simply an indicator that the use of human predictors in SDMs is species specific. It is also possible that human predictors play more important roles in modeling species distributions than climate (similar to your example with Naves et al. 2003). Thus, tests that can follow the protocol of Petitpierre et al. that use various human predictors (and also human predictors in combination with climate) could be very valuable to answer key questions about modeling human influence on species' distributions.

Thank you again for this reference. It is a great article to cite in our Discussion section. We added it to L405-407 as follows:

“Existing methods for testing the utility, importance, and performance of environmental predictors in SDMs⁸³⁻⁸⁷ can be expanded to include human predictors.”

3. Novelty Concern #3

e.g., regarding issues of projecting to areas beyond the collected data, the issue of missing the appropriate predictor variables (such as anthropogenic ones),...

- Miguel B. Araújo et al. 2019 Standards for distribution models in biodiversity assessments. *Sci. Adv.* 5, eaat4858(2019). DOI:10.1126/sciadv.aat4858

- Arenas-Castro, S., Regos, A., Martins, I., Honrado, J. & Alonso, J. (2022). Effects of input data sources on species distribution model predictions across species with different distributional ranges. *Journal of Biogeography*, 49, 1299–1312. <https://doi.org/10.1111/jbi.14382>

- Dormann, C.F., Schymanski, S.J., Cabral, J., Chuine, I., Graham, C., Hartig, F., Kearney, M., Morin, X., Römermann, C., Schröder, B. and Singer, A. (2012), Correlation and process in species distribution models: bridging a dichotomy. *Journal of Biogeography*, 39: 2119-2131. <https://doi.org/10.1111/j.1365-2699.2011.02659.x>

Per your previous review comments above (Concern #18), we followed your suggestion to delete L455-459 from our manuscript where those issues were originally discussed.

In our original manuscript, we cited Araújo et al. 2019 in L391-393 as follows:

“Recent literature has called for standardizing SDM methods^{4,79,80}, but none specifically concerning human influence.”

In our revised manuscript, we cited the Arenas-Castro et al. and Dormann articles as references in L405-407:

“Existing methods for testing the utility, importance, and performance of environmental predictors in SDMs⁸³⁻⁸⁷ can be expanded to include human predictors.”

4. Novelty Concern #4

e.g., dealing with correlating predictor variables (such as worldclim data):

-Dormann, C.F., Elith, J., Bacher, S., Buchmann, C., Carl, G., Carré, G., Marquéz, J.R.G., Gruber, B., Lafourcade, B., Leitão, P.J., Münkemüller, T., McClean, C., Osborne, P.E., Reineking, B., Schröder, B., Skidmore, A.K., Zurell, D. and Lautenbach, S. (2013), Collinearity: a review of methods to deal with it and a simulation study evaluating their performance. *Ecography*, 36: 27-46. <https://doi.org/10.1111/j.1600-0587.2012.07348.x>

We addressed this concern by removing the question on correlated predictors from Box1C, following your suggestion from Concern #15 above.

We still found this overview on correlating predictors interesting and cited it alongside the other articles as a reference for interested readers (L405-407):

“Existing methods for testing the utility, importance, and performance of environmental predictors in SDMs⁸³⁻⁸⁷ can be expanded to include human predictors.”

5. Novelty Concern #5

There is a nice summary of these issues addressed here, and further literature examples cited below can be taken from it:

Charlène, G., Bruno, D. & Thomas, S. Selecting environmental descriptors is critical for modelling the distribution of Antarctic benthic species. *Polar Biol* 43, 1363–1381 (2020).

<https://doi.org/10.1007/s00300-020-02714-2>

Calibration is a critical step in SDM procedures, influencing their relevance, robustness and accuracy (Barbet-Massin et al. 2012; Guisan et al. 2013; Anderson et al. 2016). The selection of environmental descriptors is also important, as it shapes model accuracy and performance (Elith and Leathwick 2009; Austin and van Niel 2011; Dormann et al. 2012; Braunschweig et al. 2013; Bucklin et al. 2015; Bradie and Leung 2017; Petitpierre et al. 2017). The inappropriate selection of descriptors has been shown to cause overfitting in SDMs, especially when the number of descriptors is high compared to the number of occurrences available (Anderson and Gonzalez 2011; Synes and Osborne 2011; Braunschweig et al. 2013; Kramer-Schadt et al. 2013; Petitpierre et al. 2017), leading to over-complex models, reduced transferability performances and underestimation of predicted suitable areas (Beaumont et al. 2005).

Thank you for this article. We found it to be similar to the other topics that you have mentioned which we have addressed, so we cited it in L405-407 as a reference for readers as follows:

“Existing methods for testing the utility, importance, and performance of environmental predictors in SDMs⁸³⁻⁸⁷ can be expanded to include human predictors.”

Regarding this paragraph from Charlène et al., we agree that calibration, predictor selection, overfitting, and issues with transferability are existing concerns for SDMs. Our manuscript surely covers the theme of predictor selection, however, unlike Charlène et al, our we focus specifically on human predictors as opposed to environmental predictors. As shown from the 1,429 SDM articles we found in our search among over 12,800 SDM articles, the exploration of human predictors and their performance in SDMs is relatively minimal in the literature. In this context, our synthesis and analysis are novel contributions to the field. Regarding Charlène et al.'s description of calibration and overfitting, we do not raise these issues in our manuscript, so we find no conflict in terms of novelty. Finally, regarding transferability, you had suggested that we remove a related question about transferability from Box 1C (L314 of the original manuscript), which we had done (see Concern #15).

Thank you again for your help in improving our manuscript!

Decision Letter, first revision:

7th March 2024

Dear Dr. Frans,

Thank you for submitting your revised manuscript "Gaps and opportunities in modeling human influence on species distributions in the Anthropocene" (NATECOLEVOL-23102331A). It has now been seen again by the original reviewers and their comments are below. The reviewers find that the paper has improved in revision, and therefore we'll be happy in principle to publish it in Nature Ecology & Evolution, pending minor revisions to satisfy the reviewers' final requests and to comply with our editorial and formatting guidelines.

[REDACTED]

Reviewer #1 (Remarks to the Author):

Than authors have properly responded my comments. I have no more concerns.

Canran Liu

Reviewer #2 (Remarks to the Author):

This is the second revision of the corresponding manuscript, and I would like to congratulate the authors for their very thorough and thoughtful revision. It was a pleasure to read the rebuttal letter, and the new figures have really added to the understanding of the data behind the metaanalysis. I just have (still) one minor remark: on my screen at least I cannot see the light green colour in Fig. 3 A. Please think of revising (again, sorry).

25Our ref: NATECOLEVOL-23102331A

21st March 2024

Dear Dr. Frans,

Thank you for your patience as we've prepared the guidelines for final submission of your Nature Ecology & Evolution manuscript, "Gaps and opportunities in modeling human influence on species distributions in the Anthropocene" (NATECOLEVOL-23102331A). Please carefully follow the step-by-step instructions provided in the attached file, and add a response in each row of the table to indicate the changes that you have made. Please also check and comment on any additional marked-up edits we have proposed within the text. Ensuring that each point is addressed will help to ensure that your revised manuscript can be swiftly handed over to our production team.

****We would like to start working on your revised paper, with all of the requested files and forms, as soon as possible (preferably within two weeks). Please get in contact with us immediately if you anticipate it taking more than two weeks to submit these revised files.****

In recognition of the time and expertise our reviewers provide to Nature Ecology & Evolution's editorial process, we would like to formally acknowledge their contribution to the external peer review of your manuscript entitled "Gaps and opportunities in modeling human influence on species distributions in the Anthropocene". For those reviewers who give their assent, we will be publishing their names alongside the published article.

Nature Ecology & Evolution offers a Transparent Peer Review option for new original research manuscripts submitted after December 1st, 2019. As part of this initiative, we encourage our authors to support increased transparency into the peer review process by agreeing to have the reviewer comments, author rebuttal letters, and editorial decision letters published as a Supplementary item. When you submit your final files please clearly state in your cover letter whether or not you would like to participate in this initiative. Please note that failure to state your preference will result in delays in accepting your manuscript for publication.

26Cover suggestions

We welcome submissions of artwork for consideration for our cover. For more information, please see our guide for cover artwork.

Nature Ecology & Evolution has now transitioned to a unified Rights Collection system which will allow our Author Services team to quickly and easily collect the rights and permissions required to publish your work. Approximately 10 days after your paper is formally accepted, you will receive an email in providing you with a link to complete the grant of rights. If your paper is eligible for Open Access, our Author Services team will also be in touch regarding any additional information that may be required to arrange payment for your article.

Please note that *Nature Ecology & Evolution* is a Transformative Journal (TJ). Authors may publish their research with us through the traditional subscription access route or make their paper immediately open access through payment of an article-processing charge (APC). Authors will not be required to make a final decision about access to their article until it has been accepted. Find out more about Transformative Journals

Authors may need to take specific actions to achieve compliance with funder and institutional open access mandates. If your research is supported by a funder that requires immediate open access (e.g. according to Plan S principles) then you should select the gold OA route, and we will direct you to the compliant route where possible. For authors selecting the subscription publication route, the journal's standard licensing terms will need to be accepted, including <https://www.nature.com/nature-portfolio/editorial-policies/self-archiving-and-license-to-publish>. Those licensing terms will supersede any other terms that the author or any third party may assert apply to any version of the manuscript.

Please use the following link for uploading these materials:
[REDACTED]

27[REDACTED]

Reviewer #1:

Remarks to the Author:

Than authors have properly responded my comments. I have no more concerns.

Canran Liu

Reviewer #2:

Remarks to the Author:

This is the second revision of the corresponding manuscript, and I would like to congratulate the authors for their very thorough and thoughtful revision. It was a pleasure to read the rebuttal letter, and the new figures have really added to the understanding of the data behind the metaanalysis. I just have (still) one minor remark: on my screen at least I cannot see the light green colour in Fig. 3 A. Please think of revising (again, sorry).

Author Rebuttal, first revision:

Response to Reviewers

(Reviewers' Comments in bold font and Authors' Responses in regular font)

NATECOLEVOL-23102331A

“Gaps and opportunities in modeling human influence on species distributions in the Anthropocene”

Reviewer #1 (Remarks to the Author):

Remarks to the Author:

Than authors have properly responded my comments. I have no more concerns.

Canran Liu

Dear Dr. Liu,

28We are so grateful for your approval of our manuscript. Thank you for reviewing our paper a second time. We appreciated your insights!

Sincerely,

Veronica Frans and Jianguo (Jack) Liu

Reviewer #2 (Remarks to the Author):

This is the second revision of the corresponding manuscript, and I would like to congratulate the authors for their very thorough and thoughtful revision. It was a pleasure to read the rebuttal letter, and the new figures have really added to the understanding of the data behind the metaanalysis.

I just have (still) one minor remark: on my screen at least I cannot see the light green colour in Fig. 3 A. Please think of revising (again, sorry).

Dear Reviewer #2,

We are so happy that you approve of our revisions! These revisions are thanks to you, so thank you so much for your insights! Please don't worry about this last remark. We tried looking at Fig. 3A on different screens and noticed the difficulty you must have experienced. Our apologies for that, but thank you for letting us know. We changed the color gradient of this figure to much stronger colors (purple to blue to green to orange), so there shouldn't be any more issues.

Thank you again for your great help and care!

Sincerely,

Veronica Frans and Jianguo (Jack) Liu

Final Decision Letter:

25th April 2024

Dear Ms Frans,

We are pleased to inform you that your Analysis entitled "Gaps and opportunities in modeling human influence on species distributions in the Anthropocene", has now been accepted for publication in Nature Ecology & Evolution.

29Over the next few weeks, your paper will be copyedited to ensure that it conforms to Nature Ecology and Evolution style. Once your paper is typeset, you will receive an email with a link to choose the appropriate publishing options for your paper and our Author Services team will be in touch regarding any additional information that may be required

Due to the importance of these deadlines, we ask you please us know now whether you will be difficult to contact over the next month. If this is the case, we ask you provide us with the contact information (email, phone and fax) of someone who will be able to check the proofs on your behalf, and who will be available to address any last-minute problems . Once your paper has been scheduled for online publication, the Nature press office will be in touch to confirm the details.

Acceptance of your manuscript is conditional on all authors' agreement with our publication policies (see www.nature.com/authors/policies/index.html). In particular your manuscript must not be published elsewhere and there must be no announcement of the work to any media outlet until the publication date (the day on which it is uploaded onto our web site).

Please note that *Nature Ecology & Evolution* is a Transformative Journal (TJ). Authors may publish their research with us through the traditional subscription access route or make their paper immediately open access through payment of an article-processing charge (APC). Authors will not be required to make a final decision about access to their article until it has been accepted. Find out more about Transformative Journals

Authors may need to take specific actions to achieve compliance with funder and institutional open access mandates. If your research is supported by a funder that requires immediate open access (e.g. according to Plan S principles) then you should select the gold OA route, and we will direct you to the compliant route where possible. For authors selecting the subscription publication route, the journal's standard licensing terms will need to be accepted, including <https://www.nature.com/nature-portfolio/editorial-policies/self-archiving-and-license-to-publish>. Those licensing terms will supersede any other terms that the author or any third party may assert apply to any version of the manuscript.

30We welcome the submission of potential cover material (including a short caption of around 40 words) related to your manuscript; suggestions should be sent to Nature Ecology & Evolution as electronic files (the image should be 300 dpi at 210 x 297 mm in either TIFF or JPEG format). Please note that such pictures should be selected more for their aesthetic appeal than for their scientific content, and that colour images work better than black and white or grayscale images. Please do not try to design a cover with the Nature Ecology & Evolution logo etc., and please do not submit composites of images related to your work. I am sure you will understand that we cannot make any promise as to whether any of your suggestions might be selected for the cover of the journal.

You can generate the link yourself when you receive your article DOI by entering it here: <http://authors.springernature.com/share>.

[REDACTED]

P.S. Click on the following link if you would like to recommend Nature Ecology & Evolution to your librarian <http://www.nature.com/subscriptions/recommend.html#forms>

** Visit the Springer Nature Editorial and Publishing website at www.springernature.com/editorial-and-publishing-jobs for more information about our career opportunities. If you have any questions please click here.**